# Model LEGO: Creating Models Like Disassembling and Assembling Building Blocks

**Jiacong Hu**[1,5], **Jing Gao**[2], **Jingwen Ye**[3],
**Yang Gao**[7], **Xingen Wang**[1,7], **Zunlei Feng**[4,5,6 *], **Mingli Song**[1,5,6]

[1]College of Computer Science and Technology, Zhejiang University,
[2] Robotics Institute, School of Computer Science, Carnegie Mellon University,
[3]Electrical and Computer Engineering, National University of Singapore
[4]School of Software Technology, Zhejiang University,
[5]State Key Laboratory of Blockchain and Data Security, Zhejiang University,
[6]Hangzhou High-Tech Zone (Binjiang) Institute of Blockchain and Data Security,
[7]Bangsheng Technology Co., Ltd.

jiaconghu@zju.edu.cn,jinggao2@andrew.cmu.edu,jingweny@nus.edu.sg,
{roygao,newroot,zunleifeng,brooksong}@zju.edu.cn

## Abstract

With the rapid development of deep learning, the increasing complexity and scale of parameters make training a new model increasingly resource-intensive. In this paper, we start from the classic convolutional neural network (CNN) and explore a paradigm that does not require training to obtain new models. Similar to the birth of CNN inspired by receptive fields in the biological visual system, we draw inspiration from the information subsystem pathways in the biological visual system and propose Model Disassembling and Assembling (MDA). During model disassembling, we introduce the concept of relative contribution and propose a component locating technique to extract task-aware components from trained CNN classifiers. For model assembling, we present the alignment padding strategy and parameter scaling strategy to construct a new model tailored for a specific task, utilizing the disassembled task-aware components. The entire process is akin to playing with LEGO bricks, enabling arbitrary assembly of new models, and providing a novel perspective for model creation and reuse. Extensive experiments showcase that task-aware components disassembled from CNN classifiers or new models assembled using these components closely match or even surpass the performance of the baseline, demonstrating its promising results for model reuse. Furthermore, MDA exhibits diverse potential applications, with comprehensive experiments exploring model decision route analysis, model compression, knowledge distillation, and more. For more information, please visit https://model-lego.github.io/.

## 1 Introduction

Convolutional Neural Networks (CNNs), as the predominant architecture in deep learning, play a crucial role in image, video, and audio processing [1, 2, 3]. CNNs were originally inspired by the concept of receptive fields in the biological visual system [4], and our focus is to explore and leverage similar characteristics within CNNs. Various studies [5, 6, 7] have delved into unraveling the intricacies of biological visual information processing systems. Notably, Livingstone et al. [6] substantiated that the intermediate visual cortex comprises relatively independent subdivisions.

---

* Corresponding author.

38th Conference on Neural Information Processing Systems (NeurIPS 2024).

Some studies [8, 9, 10, 11, 12, 13, 14] have endeavored to visualize critical layers and neurons within CNNs. These visualizations demonstrate that in the more shallow layers of CNNs, form, color, texture, and edge features are processed by distinct convolutional kernels, whereas deeper layers are responsible for category-aware features. The above phenomena of feature processing within CNNs demonstrates the similarity to the parallel processing mechanism [5, 6] and the information integration theory [7] proposed in biological visual systems. Consequently, this paper is dedicated to the exploration and practical utilization of these distinct subsystems within CNNs.

In pursuit of this exploration, we introduce a pioneering task named Model Disassembling and Assembling (MDA), a novel approach to construct and combine subsystems with LEGO-like flexibility. The conceptual framework behind MDA posits that, much like assembling and disassembling LEGO structures, deep learning models can undergo such operations without incurring significant training overhead or compromising performance. This task is designed to be universally applicable, spanning various existing Deep Neural Networks (DNNs), such as Convolutional Neural Networks (CNNs), Graph Neural Networks (GNNs), Transformers, and others.

However, constructing the MDA framework presents several challenges, notably in determining the minimal disassembling unit and devising an assembly process with minimal impact on performance. Existing works either require the predefinition of task units during the initial training stage, or the disassembled unit cannot be directly used for inference or assembly [15, 16, 17]. In contrast, our approach distinguishes itself as the inaugural attempt to directly disassemble a trained network into task-aware components, avoiding the need for additional networks or fine-tuning. This ensures efficiency and interpretability throughout the disassembling and assembling processes.

In this paper, we illustrate our approach using CNN classifiers, with the assurance that its applicability extends to other Deep Neural Network (DNN) architectures. In the disassembling phase, we define the task-aware component by introducing the concept of relative contribution and a mechanism for contribution aggregation and allocation. This is seamlessly applied throughout the forward propagation process of the network. Building on these principles, we introduce a component locating technique that discerns and extracts task-aware components. In the assembling phase, we propose a simple yet effective alignment padding strategy. This involves padding empty kernels onto each convolutional filter to ensure uniform kernel counts across all filters in each layer. Additionally, to account for varying feature magnitudes across different components, we implement a parameter scaling strategy. The resulting MDA framework facilitates recombination among different pre-trained models, providing a vital technique for model reuse.

Our contributions in this study are summarized as follows:

- We introduce a novel task, MDA, which aims to disassemble and assemble deep models in an interpretable manner reminiscent of playing with LEGOs. This task is motivated by the subdivision of the biological visual system [6].

- We present the inaugural method for MDA, specifically applied to CNN classifiers. In the model disassembling phase, we introduce a component locating technique to disassemble task-aware components from the models. For model assembling, we propose an alignment padding strategy and a parameter scaling strategy to assemble task-aware components into a new model.

- Extensive experiments validate the efficacy of our proposed MDA method, showing that the performance of the disassembled and assembled models closely matches or even surpasses that of the baseline models.

- MDA introduces a fresh perspective for model reuse. Additionally, we explore diverse applications of MDA, including decision route analysis, model compression, knowledge distillation, and more.

## 2 Model Disassembling and Assembling

In this section, we present the definition of Model Disassembling and Assembling (MDA). Let us consider a set of $N$ pre-trained deep learning models denoted as $\{\mathcal{M}^{(n)}\}_{n=1}^{N}$. Each model $\mathcal{M}^{(n)}$ comprises $K^{(n)}$ subtasks, represented as $\{t_k^{(n)}\}_{k=1}^{K^{(n)}}$. Our objective in model disassembling is to extract the model components $\mathcal{M}[t_k^{(n)}]$ corresponding to a specific subtask $t_k^{(n)}$ from the source

model $\mathcal{M}^{(n)}$. These extracted components $\mathcal{M}[t_k^{(n)}]$ are intended to encapsulate only the parameters critically relevant to the subtask $t_k^{(n)}$. In essence, the disassembled components $\mathcal{M}[t_k^{(n)}]$ should function as an independent model, preserving the full capability for the subtask $t_k^{(n)}$ without redundant capability for other subtasks. Furthermore, the assembling process involves combining disassembled components from different models. For example, combining components $\mathcal{M}[t_{k_1}^{(n_1)}, \ldots, t_{k_2}^{(n_1)}]$ from $\mathcal{M}^{(n_1)}$ and $\mathcal{M}[t_{k_3}^{(n_2)}, \ldots, t_{k_4}^{(n_2)}]$ from $\mathcal{M}^{(n_2)}$ results in a new model $\mathcal{M}^{(new)}$. This assembled model $\mathcal{M}^{(new)} = \mathcal{M}[t_{k_1}^{(n_1)}, \ldots, t_{k_2}^{(n_1)}, t_{k_3}^{(n_2)}, \ldots, t_{k_4}^{(n_2)}]$ is expected to retain full functionality for the subtasks $\{t_{k_1}^{(n_1)}, \ldots, t_{k_2}^{(n_1)}\}$ and $\{t_{k_3}^{(n_2)}, \ldots, t_{k_4}^{(n_2)}\}$.

MDA is applicable to various existing deep learning architectures, including Convolutional Neural Networks (CNNs), Graph Neural Networks (GNNs), and Transformers. The focus of this paper is to explore the implementation and efficacy of MDA specifically in the context of CNN classifiers.

## 3 Model Disassembling

### 3.1 Contribution Aggregation and Allocation

Given a CNN classifier with $K$ categories, each category is treated as an individual subtask within the classifier. Conventionally, a Softmax operation is employed on the output features of the final fully connected layer, transforming these features into a probabilistic distribution that sums to unity. The feature with a relatively higher value corresponds to a higher probability, and the category with the highest probability serves as the predicted result of the classifier. Consequently, the feature exhibiting relatively greater magnitude plays a decisive role in determining the final classification result. We term the relative degree to which features influence the result as *relative contribution*. It is crucial to note that the concept of relative contribution is not confined solely to the final layer of the network. Features from preceding layers play a pivotal role in shaping the features of subsequent layers. Consequently, we

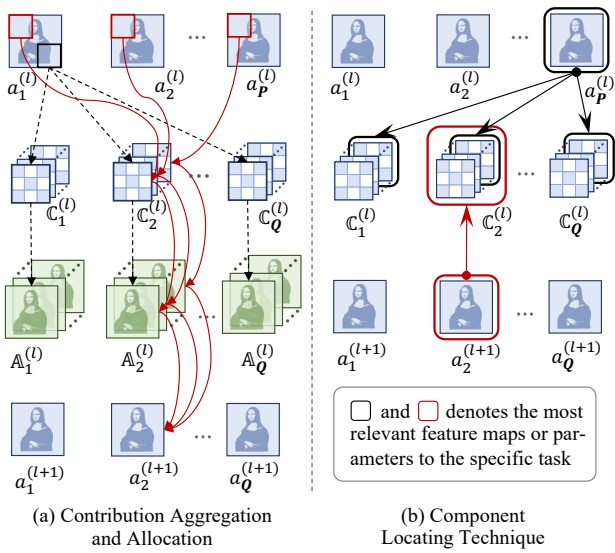

(a) Contribution Aggregation and Allocation

(b) Component Locating Technique

Figure 1: Disassembling process at the $l$-th layer of a CNN model, where the red solid line represents the contribution aggregation process, and the black dashed line represents the contribution allocation process.

extend the concept of relative contribution to encompass all layers of the CNN classifier, thereby establishing a comprehensive contribution system that spans the entirety of the network.

To illustrate this process more specifically, we focus on the $l$-th convolutional layer of a CNN (the case of the fully connected layer is discussed in Appendix A). The $l$-th layer has $\boldsymbol{P}$ input channels and $\boldsymbol{Q}$ output channels. Correspondingly, the $l$-th layer comprises $\boldsymbol{Q}$ convolution filters, denoted as $\{\mathbb{C}_q^{(l)}\}_{q=1}^{\boldsymbol{Q}}$. Each filter $\mathbb{C}_q^{(l)}$ consists of $\boldsymbol{P}$ convolution kernels, thus $\mathbb{C}_q^{(l)} = \{c_{q,p}^{(l)}\}_{p=1}^{\boldsymbol{P}}$. The input feature maps for the $l$-th layer are represented as $\{a_p^{(l)}\}_{p=1}^{\boldsymbol{P}}$. With the convolution filter $\mathbb{C}_q^{(l)}$, the hidden feature maps $\mathbb{A}_q^{(l)}$ for the $q$-th channel in the $l$-th layer are computed as follows:

$$\mathbb{A}_q^{(l)} = \{a_{q,p}^{(l)}\}_{p=1}^{\boldsymbol{P}}, \, a_{q,p}^{(l)} = c_{q,p}^{(l)} \otimes a_p^{(l)}, \tag{1}$$

where $\otimes$ denotes the convolution operation. From these hidden feature maps $\mathbb{A}_q^{(l)}$, the output feature map $a_q^{(l+1)}$ in the $l$-th layer (which serves as the input feature map $a_q^{(l+1)}$ in the $(l+1)$-th layer) is

determined as:

$$a_q^{(l+1)} = \sum_{p=1}^{P} a_{q,p}^{(l)} + b_q^{(l)}, \tag{2}$$

where $b_q^{(l)}$ is the bias for the $q$-th channel. From Eqn.(2), it is evident that the output feature map $a_q^{(l+1)}$ is the summation of the hidden feature maps $\mathbb{A}_q^{(l)}$. This implies that the value of $a_q^{(l+1)}$ is influenced by the value of each individual hidden feature map $a_{q,p}^{(l)}$. Similar to the Softmax operation in the final layer, the larger the individual hidden feature map, the greater its contribution to the output $a_q^{(l+1)}$.

### 3.1.1 Contribution Aggregation

The red solid line in Fig. 1 represents the contribution aggregation process. In this process, the contributions from the input feature maps $\{a_p^{(l)}\}_{p=1}^P$ are aggregated to the hidden feature maps $\mathbb{A}_q^{(l)}$ via the convolution filter $\mathbb{C}_q^{(l)}$ of the $q$-th channel. To quantify the contribution of each hidden feature map $a_{q,p}^{(l)}$ to the output $a_q^{(l+1)}$, we introduce a metric $s_{q,p}^{(l)}$ defined as follows:

$$s_{q,p}^{(l)} = \sum_{h=1}^{H^{(l)}} \sum_{w=1}^{W^{(l)}} a_{q,p}^{(l)}[h,w], \tag{3}$$

where $H^{(l)}$ and $W^{(l)}$ denote the height and width, respectively, of the feature map $a_{q,p}^{(l)}$. The term $a_{q,p}^{(l)}[h,w]$ represents the pixel value at the $h$-th row and $w$-th column of $a_{q,p}^{(l)}$. Considering the presence of activation functions such as ReLU, negative contributions are treated as having zero impact on the result. Consequently, the contribution $\hat{s}_{q,p}^{(l)}$ of the hidden feature map $a_{q,p}^{(l)}$ is recalculated as:

$$\hat{s}_{q,p}^{(l)} = \max(s_{q,p}^{(l)}, 0). \tag{4}$$

In line with the principle of the Softmax operation, the contribution of each hidden feature map is relative. Hence, we employ min-max normalization to obtain the relative contribution $r_{q,p}^{(l)}$ of each hidden feature map $a_{q,p}^{(l)}$:

$$r_{q,p}^{(l)} = \frac{\hat{s}_{q,p}^{(l)} - \min(\{\hat{s}_{q,p}^{(l)}\}_{p=1}^P)}{\max(\{\hat{s}_{q,p}^{(l)}\}_{p=1}^P) - \min(\{\hat{s}_{q,p}^{(l)}\}_{p=1}^P) + \varepsilon}, \tag{5}$$

where $\varepsilon$ is a small constant added to prevent division by zero. This normalization process ensures that the contributions are scaled relative to each other, facilitating the identification of the most influential hidden feature maps in the layer.

### 3.1.2 Contribution Allocation

The black dashed line in Fig. 1 represents the process of contribution allocation, where the contribution from the feature map $a_p^{(l)}$ is allocated to various hidden feature maps $\{a_{q,p}^{(l)}\}_{q=1}^Q$ through different convolution kernels $\{c_{q,p}^{(l)}\}_{q=1}^Q$. Consequently, the overall contribution $s_p^{(l)}$ of the feature map $a_p^{(l)}$ is calculated using the following equation:

$$s_p^{(l)} = \sum_{q=1}^{Q} r_{q,p}^{(l)}. \tag{6}$$

In this calculation, similar to the previous steps, negative contributions are considered as zero:

$$\hat{s}_p^{(l)} = \max(s_p^{(l)}, 0). \tag{7}$$

Moreover, acknowledging that the significance of each feature map is relative, the relative contribution $r_p^{(l)}$ of the feature map $a_p^{(l)}$ is computed as follows:

$$r_p^{(l)} = \frac{\hat{s}_p^{(l)} - \min(\{\hat{s}_p^{(l)}\}_{p=1}^P)}{\max(\{s_p^{(l)}\}_{p=1}^P) - \min(\{\hat{s}_p^{(l)}\}_{p=1}^P) + \varepsilon}. \tag{8}$$

## 3.2 Component Locating Technique

Building upon the concept of relative contribution and the mechanism of contribution aggregation and allocation, we introduce a novel approach termed the component locating technique. This method is designed to identify task-aware components, that is, the parameters most relevant to a given task, within a CNN. Taking the $l$-th layer as an example, the initial stage of this technique involves identifying the most relevant feature maps for the targeted task. Subsequently, the next step is to pinpoint the most relevant parameters by discerning which parameters are linked to the identified most relevant feature maps. For a specific task, please refer to Appendix B.

### 3.2.1 Relevant Feature Identifying

In essence, the most relevant feature maps are those which exhibit a relatively larger contribution to the result of the model. In the process of contribution aggregation, a threshold value denoted as $\alpha$ is employed to discern whether a relative contribution $r_{q,p}^{(l)}$ (calculated in Eqn. 5) is large or small:

$$\hat{r}_{q,p}^{(l)} = \begin{cases} 1 & r_{q,p}^{(l)} \geq \alpha \\ 0 & r_{q,p}^{(l)} < \alpha \end{cases}, \tag{9}$$

where $\alpha$ is a chosen value within the range (0,1]. Through the above equation, the soft relative contribution $r_{q,p}^{(l)}$, which are continuous values indicating the degree of contribution, are transformed into hard relative contribution $\hat{r}_{q,p}^{(l)}$. The $s_p^{(l)}$ in Eqn. 6 is now the sum over the hard relative contribution $\hat{r}_{q,p}^{(l)}$. Eqn.7 and Eqn.8 remain unchanged.

Further, a second threshold value $\beta$ is used to determine whether the relative contribution $r_p^{(l)}$ (as defined in Eqn.8) of a feature map is large or small:

$$\hat{r}_p^{(l)} = \begin{cases} 1 & r_p^{(l)} \geq \beta \\ 0 & r_p^{(l)} < \beta \end{cases}, \tag{10}$$

where $\beta$ is a chosen value within the range (0,1]. This equation transforms the soft relative contribution $r_p^{(l)}$ into hard relative contribution $\hat{r}_p^{(l)}$, just as in the case of $\hat{r}_{q,p}^{(l)}$.

### 3.2.2 Parameter Linking

In Eqn. 10, the hard relative contribution $\hat{r}_p^{(l)}$, being either 0 or 1, indicates whether the feature map $a_p^{(l)}$ is most relevant to the predicted result. Similarly, the hard relative contribution $\hat{r}_q^{(l+1)}$ can also reflect whether the feature map $a_q^{(l+1)}$ is most relevant to the predicted result. Integrating Eqn.1 with Eqn.2, we can represent the convolutional process as follows:

$$a_q^{(l+1)} = \sum_{p=1}^{P} c_{q,p}^{(l)} \otimes a_p^{(l)} + b_q^{(l)}, \tag{11}$$

This equation signifies that the output feature map $a_q^{(l+1)}$ is generated by the convolution operation of the input feature maps $\{a_p^{(l)}\}_{p=1}^{P}$ with the respective convolution kernels $\{c_{q,p}^{(l)}\}_{p=1}^{P}$, which collectively form the convolution filter $\mathbb{C}_q^{(l)}$. Thus, if the output feature map $a_q^{(l+1)}$ is determined to be most relevant to the predicted result, then it logically follows that the convolution filter $\mathbb{C}_q^{(l)}$ and the associated bias $b_q^{(l)}$ are also most relevant to the predicted result. Furthermore, each input feature map $a_p^{(l)}$ engages in the convolution operation exclusively with the kernels $\{c_{q,p}^{(l)}\}_{q=1}^{Q}$ across different convolution filters. Therefore, if the feature map $a_p^{(l)}$ is identified as most relevant to the predicted result, then the convolution kernels $\{c_{q,p}^{(l)}\}_{q=1}^{Q}$ associated with it are also deemed most relevant to the predicted result. The component locating technique is also applicable to fully connected layers, enabling the identification of the most relevant filters, kernels, and biases.

In summary, through the component locating technique, we can effectively identify and discriminate the most relevant components to a specific task. This includes pinpointing the most relevant filters, kernels, and biases. Subsequently, the identified parameters most relevant to a specific task can be extracted from the source model through a process known as structure pruning [18].

## 4 Model Assembling

In the process of CNN model assembling, our objective is to combine disassembled, task-aware components, typically derived from different source models, into a new model. This assembling is performed without necessitating retraining or incurring performance loss. To achieve this goal, we propose an alignment padding strategy and a parameter scaling strategy. It is important to note that the focus of this paper is specifically on assembling different models that share isomorphic network architectures.

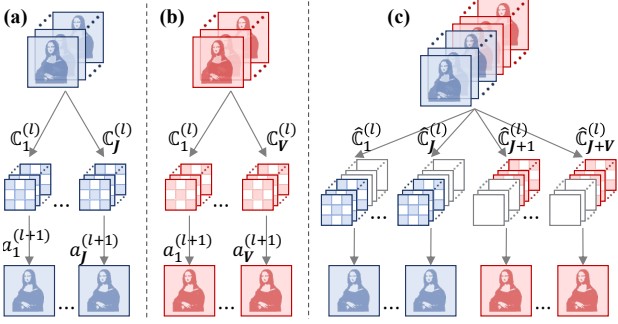

Figure 2: Assembling process at the $l$-th layer of CNN models: **(a)** and **(b)** represent two distinct disassembled models, respectively; **(c)** illustrates the assembled model.

### 4.1 Alignment Padding Strategy

In the assembling of CNN models, we combine models layer by layer along the dimension of the filters. A challenge in this process is that filters from different models may have varying numbers of kernels. To address this, we introduce a simple yet effective alignment padding strategy. This strategy involves padding empty kernels to each filter, ensuring that all filters in a given layer have a uniform number of kernels.

Consider the $l$-th layer as an example. As depicted in Fig.2(a), a disassembled model consists of $J$ convolution filters, denoted as $\{\mathbb{C}_j^{(l)}\}_{j=1}^J$. Each convolution filter $\mathbb{C}_j^{(l)}$ comprises $I$ convolutional kernels, expressed as $\mathbb{C}_j^{(l)} = \{c_{j,i}^{(l)}\}_{i=1}^I$. Another disassembled model, shown in Fig.2(b), contains $V$ convolution filters $\{\mathbb{C}_v^{(l)}\}_{v=1}^V$, with each filter $\mathbb{C}_v^{(l)}$ consisting of $U$ convolutional kernels, expressed as $\mathbb{C}_v^{(l)} = \{c_{v,u}^{(l)}\}_{u=1}^U$. In the assembling process, illustrated in Fig.2(c), these two models are merged into a new model with $J + V$ convolution filters. Each filter in this new model is augmented with a complementary number of empty kernels. Specifically, if a convolution filter $\hat{\mathbb{C}}_j^{(l)}$ originates from $\mathbb{C}_j^{(l)}$, it is padded with $U$ empty kernels at the end, forming $\hat{\mathbb{C}}_j^{(l)} = \{c_{j,1}^{(l)}, c_{j,2}^{(l)}, \ldots, c_{j,I}^{(l)}, 0_1, 0_2, \ldots, 0_U\}$. Conversely, if a convolution filter $\hat{\mathbb{C}}_v^{(l)}$ comes from $\mathbb{C}_v^{(l)}$, it is padded with $I$ empty kernels at the beginning, resulting in $\hat{\mathbb{C}}_v^{(l)} = \{0_1, 0_2, \ldots, 0_I, c_{v,1}^{(l)}, c_{v,2}^{(l)}, \ldots, c_{v,U}^{(l)}\}$. Through this alignment padding strategy, every convolution filter in the assembled model is standardized to have the same total of $I + U$ convolution kernels.

The alignment padding strategy can be readily extended to incorporate multiple disassembled models and applied across all layers of a CNN. A key feature of this approach is that the assembled model is ready for inference immediately, without the need for any retraining.

### 4.2 Parameter Scaling Strategy

In addition to the alignment padding strategy, we address another critical issue in model assembling: the potential disparity in the magnitude of features output by disassembled components from different source models. If left unaddressed, this disparity could cause the assembled model to be biased towards the disassembled components with larger feature magnitudes, impacting the balance and effectiveness of the assembled model. To resolve this, we propose a parameter scaling strategy.

Taking the $l$-th layer as an example, as shown in Fig.2(a) and (b), let's consider the output feature maps in the $l$-th layer of the disassembled models, denoted as $\{a_j^{(l+1)}\}_{j=1}^J$ and $\{a_v^{(l+1)}\}_{v=1}^V$, respectively. The magnitude of each feature map $a_j^{(l+1)}$ is quantified as follows:

$$e_j^{(l+1)} = \sum_{h=1}^{H^{(l+1)}} \sum_{w=1}^{W^{(l+1)}} a_j^{(l+1)}[h, w], \tag{12}$$

Table 1: Comparison of disassembling performance. In the classifier, each category corresponds to a task. 'Base.' refers to the average accuracy for 'Disassembled Task' in the source model. In 'Disa.', '*Score1* (+*Score2*)' represents two metrics: '*Score1*' is the accuracy and '*Score2*' is the improved accuracy compared to 'Base.' for the 'Disassembled Task' in the disassembled model.

| Dataset | Disassembled Task | VGG-16 Base. (%) | VGG-16 Disa. (%) | ResNet-50 Base. (%) | ResNet-50 Disa. (%) | GoogleNet Base. (%) | GoogleNet Disa. (%) |
|---------|-------------------|------------------|------------------|---------------------|---------------------|---------------------|---------------------|
| CIFAR-10 | 0 | 94.40 | 100.00 (+*5.60*) | 96.10 | 100.00 (+*3.90*) | 95.40 | 100.00 (+*4.60*) |
| | 1 | 96.50 | 100.00 (+*3.50*) | 96.20 | 100.00 (+*3.80*) | 97.40 | 100.00 (+*2.60*) |
| | 0-2 | 93.87 | 95.47 (+*1.60*) | 94.93 | 97.43 (+*2.50*) | 94.47 | 98.17 (+*3.70*) |
| | 3-9 | 92.49 | 92.27 (-*0.22*) | 93.81 | 94.46 (+*0.65*) | 93.46 | 93.17 (-*0.29*) |
| CIFAR-100 | 0 | 84.00 | 100.00 (+*16.00*) | 92.00 | 100.00 (+*8.00*) | 90.00 | 100.00 (+*10.00*) |
| | 1 | 87.00 | 100.00 (+*13.00*) | 87.00 | 100.00 (+*13.00*) | 85.00 | 100.00 (+*15.00*) |
| | 0-19 | 71.05 | 82.50 (+*11.45*) | 75.55 | 77.15 (+*1.60*) | 75.90 | 87.55 (+*11.65*) |
| | 20-69 | 72.74 | 79.66 (+*6.92*) | 77.68 | 79.72 (+*2.04*) | 76.60 | 82.42 (+*5.82*) |
| Tiny-ImageNet | 0 | 82.00 | 100.0 (+*18.00*) | 92.00 | 100.00 (+*8.00*) | 88.00 | 100.00 (+*12.00*) |
| | 1 | 70.00 | 100.0 (+*30.00*) | 80.00 | 100.00 (+*20.00*) | 76.00 | 100.00 (+*24.00*) |
| | 0-69 | 50.17 | 55.49 (+*5.32*) | 56.06 | 56.40 (+*0.34*) | 52.89 | 59.40 (+*6.51*) |
| | 70-179 | 45.36 | 47.95 (+*2.59*) | 51.27 | 53.42 (+*2.15*) | 47.91 | 51.04 (+*3.13*) |

Similarly, the magnitude $e_v^{(l+1)}$ for feature map $a_v^{(l+1)}$ is calculated using the same equation. We then compute the average magnitude of the feature maps from all disassembled models in the $l$-th layer:

$$\bar{e}^{(l+1)} = \frac{1}{J+V}\left(\sum_{j=1}^{J} e_j^{(l+1)} + \sum_{v=1}^{V} e_v^{(l+1)}\right). \tag{13}$$

Then, the convolution filter $\mathbb{C}_j^{(l)}$ is scaled as follows:

$$\widetilde{\mathbb{C}}_j^{(l)} = (\bar{e}^{(l+1)}/e_j^{(l+1)})\mathbb{C}_j^{(l)}. \tag{14}$$

In practical applications, this parameter scaling strategy is particularly crucial in the last fully connected layer to ensure that the magnitude differences in the final outputs of disassembled models from different source models are not excessively large.

## 5 Experiments

### 5.1 Experimental Settings

**Dataset and Network.** We select three datasets and three mainstream CNN classifiers to evaluate our MDA method. The datasets include CIFAR-10 [19], CIFAR-100 [19], and Tiny-ImageNet [20]. The chosen CNN classifiers are VGG-16 [21], ResNet-50 [22], and GoogleNet [23].

**Parameter Configuration.** In our MDA method, the key parameters are $\alpha$ and $\beta$, as defined in Eqn.9 and Eqn.10, respectively. By default, we set $\alpha = 0.3$ and $\beta = 0.2$ in convolutional layers, and $\alpha = 0.4$ and $\beta = 0.3$ in fully connected layers, unless specified otherwise. The model training is conducted using the SGD optimizer, with a learning rate of $0.01$. To ensure the reliability and reproducibility of our results, we report the average of three independent experimental runs for each result. Comprehensive details and the *source code* can be found in the *Supplementary Material*.

### 5.2 MDA Applied to CNN Models

#### 5.2.1 Model Disassembling Results

We present the results of model disassembling in Table 1. The results in Table 1 reveal that both single-task and multi-task disassembling, as executed by our method (denoted as 'Disa.'), exhibit accuracies comparable to, or even surpassing, those of the source model (denoted as 'Base.'). Notably, disassembling single tasks '0' or '1' from CIFAR-10 when using GoogleNet achieved a 100% accuracy rate. Similarly, the accuracy for disassembling multiple tasks '70-169' from Tiny-ImageNet on ResNet-50 showed an improvement of over 2.15% compared to the source model. What's more, to

Table 2: Comparison of assembling performance. 'Base.' indicates the average accuracy for the 'Assembled Task' in the source models. In 'Asse.', '*Score1 / Score2*' represent the average accuracy scores for the 'Assembled Task' in the assembled models without fine-tuning and with ten epochs of fine-tuning, respectively.

| Dataset | Assembled | VGG-16 | | ResNet-50 | | GoogleNet | |
|---|---|---|---|---|---|---|---|
| | Task | Base. (%) | Asse. (%) | Base. (%) | Asse. (%) | Base. (%) | Asse. (%) |
| CIFAR-10 + CIFAR-100 | 0 + 0 | 89.20 | 87.00 / 88.43 | 94.05 | 77.30 / 87.36 | 99.60 | 94.25 / 95.27 |
| | 0-2 + 0-19 | 74.03 | 74.17 / 74.19 | 78.08 | 64.34 / 76.37 | 78.32 | 79.22 / 79.22 |
| | 3-9 + 20-69 | 75.16 | 73.72 / 74.25 | 79.66 | 72.03 / 75.25 | 78.67 | 70.24 / 76.37 |
| | 0-9 + 20-99 | 74.87 | 72.07 / 73.18 | 79.54 | 65.97 / 74.65 | 78.57 | 66.59 / 70.36 |
| CIFAR-10 + Tiny-ImageNet | 0 + 0 | 88.20 | 94.70 / 90.72 | 94.05 | 86.95 / 94.51 | 91.70 | 62.40 / 80.34 |
| | 0-2 + 0-69 | 51.97 | 53.20 / 53.20 | 57.65 | 43.09 / 56.38 | 54.59 | 57.51 / 57.81 |
| | 3-9 + 0-69 | 54.02 | 50.20 / 52.48 | 59.49 | 52.14 / 58.63 | 56.57 | 53.74 / 55.32 |
| | 0-9 + 70-179 | 49.33 | 42.30 / 47.98 | 54.85 | 47.21 / 55.17 | 51.73 | 48.00 / 52.16 |
| CIFAR-100 + Tiny-ImageNet | 0 + 0 | 83.00 | 50.00 / 76.28 | 92.00 | 53.00 / 87.34 | 89.00 | 50.00 / 85.28 |
| | 0-19 + 0-69 | 69.86 | 50.66 / 57.19 | 74.59 | 57.86 / 69.23 | 74.73 | 58.67 / 69.14 |
| | 20-69 + 70-179 | 71.48 | 50.08 / 65.71 | 76.54 | 56.53 / 69.79 | 75.08 | 54.09 / 71.27 |
| | 0-99 + 0-199 | 55.97 | 43.06 / 56.13 | 61.51 | 53.05 / 57.23 | 59.19 | 48.66 / 58.28 |

go deeper into the performance of our proposed disassembled method, we present the disassembling results on ImageNet [24] and the comparison of parameter size and Floating Point Operations Per Second (FLOPs) in the *Supplementary Material*.

### 5.2.2 Model Assembling Results

The performance of model assembling across different datasets is presented in Table 2. It is observed that the assembled models generally achieve comparable performance to the source models in both single-task and multi-task assembling settings. Notably, the assembled models combining '0-2 + 0-69' from 'CIFAR-10 + Tiny-ImageNet' on GoogleNet surpass the source model in terms of accuracy. However, there are instances, such as with '20-69 + 70-179' from 'CIFAR-100 + Tiny-ImageNet' on ResNet-50, where a decrease in accuracy is noted. This could be attributed to the interaction and interference among the numerous parameters from the different models being assembled, particularly when the number of tasks is large, leading to less stable predictions in the new model.

### 5.3 MDA Applied to GCN Model

The proposed MDA method extends beyond CNN models and is equally applicable to Graph Convolutional Network (GCN) models [25]. We demonstrate this by conducting a disassembling experiment for node classification using a GCN model [25] on the Cora dataset [26]. The results of this experiment are presented in Table 3.

The results clearly indicate that the accuracy of the disassembled GCN model surpasses that of the source model. For instance, the accuracy of the disassembled model for categories '1-2' shows an improvement of 1.47% over the source GCN model.

Table 3: Performance of the GCN model disassembling on the Cora Dataset. 'Base.' represents the average accuracy for the 'Disassembled Task' in the source model. In 'Disa.', 'Score1 (+*Score2*)' indicates the accuracy and the improvement in accuracy ('*Score2*') compared to 'Base.' for the 'Disassembled Task' of the disassembled model.

| Dataset | Disassembled | GCN | |
|---|---|---|---|
| | Task | Base. (%) | Disa. (%) |
| Cora | 0 | 72.36 | 100.00 (+*27.36*) |
| | 1 | 78.54 | 97.10 (+*18.56*) |
| | 1-2 | 88.27 | 89.74 (+*1.47*) |
| | 3-5 | 80.23 | 81.34 (+*1.11*) |

### 5.4 Ablation Study

**Parameters $\alpha$ and $\beta$.** Fig. 3 presents an ablation study on the thresholds $\alpha$ and $\beta$ in the fully connected layer and convolutional layer. With the increase of the thresholds $\alpha$ and $\beta$, fewer parameters will be regarded as relevant to the specific tasks. Therefore, in Fig. 3(a), we observe that as the

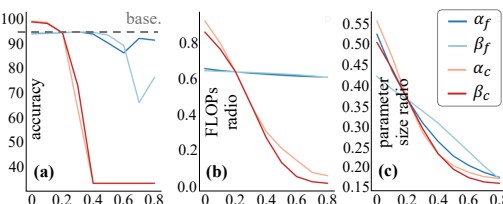

Figure 3: Accuracy curve **(a)**, FLOPs ratio curve **(b)**, and model parameter size ratio curve **(c)** for the disassembled model, varying with hyperparameters in the fully connected layers ($\alpha_f, \beta_f$) and convolutional layers ($\alpha_c, \beta_c$).

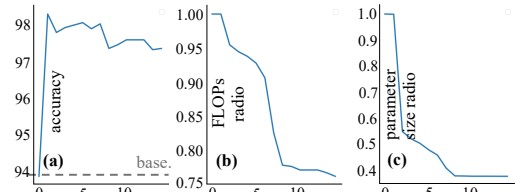

Figure 4: Accuracy curve **(a)**, FLOPs ratio curve **(b)**, and model parameter size ratio curve **(c)** for the disassembled model as the number of disassembled layers varies. The number of disassembled layers accumulates from deep layers to shallow layers in the model.

thresholds $\alpha$ and $\beta$ increase, the accuracy of the disassembled model decreases, with thresholds in the convolutional layer being more sensitive than in the fully connected layer. Fig. 3(b) illustrates that thresholds $\alpha$ and $\beta$ induce more significant changes in FLOPs in the convolutional layer compared to the fully connected layer. In Fig. 3(c), the model parameter size ratio of the disassembled model decreases with the increase in thresholds.

**Number of Disassembled Layers.** Fig. 4 shows the ablation study on the number of disassembled layers. As depicted in Fig. 4(a), the accuracy initially increases and then decreases with the rising number of disassembled layers, but consistently remains higher than the source model. Additionally, Fig. 4(b, c) reveals that both the FLOPs ratio and parameter size ratio decline as more layers are disassembled. This ablation study validates that our proposed disassembled method can effectively disassemble all layers of the model.

**Assembling Strategy.** The impact of different assembling strategies is detailed in Table 4. It is observed that the method combining padding alignment and parameter scaling ('+Padd. +Para.') results in higher accuracy compared to the strategy employing only padding alignment ('+Padd.'). In the case of '0 + 0', the '+Padd. +Para.' approach leads to a significant accuracy increase of 37.00%. These results affirm the effectiveness of the assembling strategies proposed in this paper.

Table 4: Comparison of assembling strategies. 'Base.' represents the average accuracy for the 'Assembled Task' in the source models. '+Padd.' and '+Padd. +Para.' denote the accuracy of the assembled classifier using only the padding alignment strategy and both the padding alignment and parameter scaling strategies, respectively.

| Dataset | Assembled Task | VGG-16 | | |
|---|---|---|---|---|
| | | Base. | +Padd. | +Padd. +Para. |
| CIFAR-10 + CIFAR-100 | 0 + 0 | 89.20 | 50.00 | 87.00 |
| | 0-2 + 0-19 | 74.03 | 71.74 | 74.17 |
| | 3-9 + 20-69 | 75.16 | 71.97 | 73.72 |
| | 0-9 + 20-99 | 74.87 | 69.12 | 72.07 |

# 6 Related Works

## 6.1 Model Explanation

Model explanation methods can generally be categorized into four types: activation-based, gradient-based, perturbation-based techniques, and Layer-wise Relevance Propagation (LRP). Activation-based approaches [27, 28, 29, 30, 31, 32, 33, 34] involve calculating a set of weights and then aggregation feature maps to highlight crucial features. Gradient-based methods [35, 36, 37, 38, 39, 40, 21, 41, 42] utilize gradients to identify key features. Perturbation-based techniques [43, 44, 45, 46] discern important features by altering or masking them and observing the resultant changes in output. LRP involves backward propagation of the final prediction through the network using specific local propagation rules, grounded in the principle of conservation [47]. Montavon et al. [48] provided a comprehensive review of LRP rules, including LRP-0, LRP-$\epsilon$, LRP-$\gamma$, LRP-$\alpha\beta$, flat, $\omega^2$-rules, and $z^{\mathcal{B}}$-rule, and discussed their distinctions and interconnections. Additionally, Ancona et al. [49] examined various gradient-based techniques (such as Gradient $\times$ Input, Integrated Gradients, and DeepLIFT) and LRP from both theoretical and practical angles, highlighting their similarities and

conditions for equivalence or approximation. While these model explanation techniques are primarily employed to locate important features in the original input contributing to the final prediction, our focus is distinct. We aim to identify task-aware components, namely the relevant parameters for specific tasks, for model disassembling.

## 6.2 Subnetwork Identification

Subnetwork identification approaches can be divided into ante-hoc and post-hoc techniques. Ante-hoc techniques, such as those proposed by Li et al. [50] and Liang et al. [15], incorporate novel architectural control modules to select specific filters or employ category-specific gating during training, mainly for network interpretation and adversarial sample detection. Post-hoc techniques are further subdivided. The first family requires additional learnable modules and extra training steps. For instance, Hu et al. [16] introduced Neural Architecture Disentanglement (NAD) for disentangling pre-trained DNNs into task-specific sub-architectures, while Wang et al. [51] and Frankle et al. [52] focused on data routing paths and network acceleration, respectively. Furthermore, Yu et al. [53] and Yang et al. [17] used knowledge distillation to dissect and reassemble models. The second family aligns more closely with feature attribution, with techniques such as those of Khakzar et al. [54] employing concepts similar to perturbation-based methods for pathway selection. In summary, while existing subnetwork identification techniques are commonly applied for network interpretation and adversarial sample detection, our work centers on MDA. We focus on disassembling task-aware components from trained CNN classifiers and reassembling them into a new model, akin to playing with LEGOs, without requiring additional training.

## 7 Limitation and Future Work

Our experiments, as detailed in Table 1 of the main text, demonstrate that models disassembled using the proposed method can surpass the source model in terms of accuracy. However, the performance of the assembled models, as shown in Table 2 of the main text, indicates a decrease in accuracy in certain cases (e.g., '0-19 + 0-69', '20-69 + 70-179', '0-99 + 0-199' for 'CIFAR-100 + Tiny-ImageNet'). This decline in performance could be attributed to the interference of irrelevant components, which may adversely affect the correct prediction of samples. Looking ahead, our research will concentrate on addressing the disturbance caused by irrelevant components and enhancing the effectiveness of our model disassembling and assembling technique, particularly for targeted tasks. Additionally, while this paper has focused exclusively on CNN classifiers, future research will explore the disassembling and assembling of models in other domains, including object detection and segmentation.

## 8 Conclusion

In this paper, we introduce a novel Model Disassembling and Assembling (MDA) task, inspired by the subdivision of the visual system [6], with the objective of disassembling and assembling deep models in a manner akin to playing with LEGOs. The primary focus of this paper centers on the application of MDA to CNN classifiers. During model disassembling, we introduce the concept of relative contribution and propose a component locating technique to extract task-aware components from trained CNN classifiers. For model assembling, we introduce the alignment padding strategy and parameter scaling strategy to construct a new model tailored for a specific task using the disassembled task-aware components. Extensive experiments conducted in this study reveal that the performance of the disassembled and assembled models closely aligns with or even surpasses that of the baseline models. In addition to offering a fresh perspective for model reuse, our research extends to the diverse applications of MDA, including decision route analysis, model compression, knowledge distillation, and more. In future work, we will focus on the MDA applied to other models, such as multi-modal models, large language models etc.

## Acknowledgments and Disclosure of Funding

This work is funded by National Key Research and Development Project (Grant No: 2022YFB2703100), Ningbo Natural Science Foundation (2022J182) and the Fundamental Research Funds for the Central Universities (226-2024-00145).

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

# A    Contribution Aggregation and Allocation in the Fully Connected Layer

Figure 5 illustrates the scenario of contribution aggregation and allocation in the fully connected (FC) layer within Convolutional Neural Networks (CNNs). The FC layer is composed of $Q$ filters $\{\mathbb{C}_q^{(l)}\}_{q=1}^{Q}$, each comprising $P$ kernels $\mathbb{C}_q^{(l)} = \{c_{q,p}^{(l)}\}_{p=1}^{P}$. Notably, in the FC layer, both kernels and features are single real numbers, i.e., $a_p^{(l)} \in \mathbb{R}$ and $c_{q,p}^{(l)} \in \mathbb{R}$, contrasting with the convolutional layer where kernels and features are two-dimensional matrices of real numbers, denoted as $a_p^{(l)} \in \mathbb{R}^{H \times W}$ and $c_{q,p}^{(l)} \in \mathbb{R}^{H_k^{(l)} \times W_k^{(l)}}$ (with $H^{(l)}$ and $W^{(l)}$ representing the height and width of features, and $H_k^{(l)}$ and $W_k^{(l)}$ representing the height and width of kernels).

Given this structural discrepancy, the method for computing contributions in fully connected layers requires adaptation. Specifically, the contribution $s_{q,p}^{(l)}$ defined in Eqn.(3) can be directly equated to the hidden feature $a_{q,p}^{(l)}$ as expressed below:

$$s_{q,p}^{(l)} = a_{q,p}^{(l)}. \tag{15}$$

The remaining equations in the main text remain unchanged.

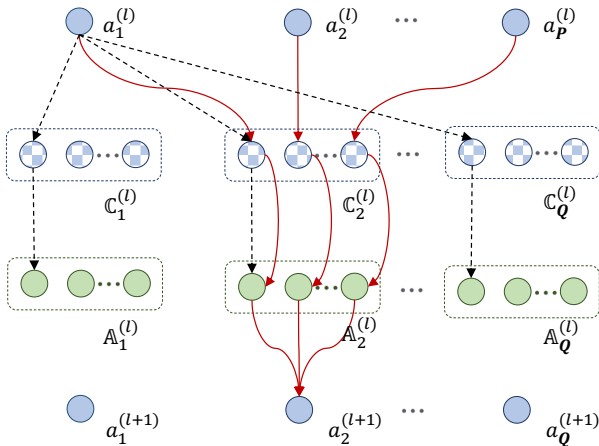

Figure 5: The process of contribution aggregation and allocation at the $l$-th fully connected layer, wherein the red solid line delineates the contribution aggregation process, and the black dashed line signifies the contribution allocation process.

# B    Component Locating Technique for a Specific Task

Figure 6 illustrates a visualization of the soft relative contribution $r_p^{(l)}$ for input feature maps in an intermediate convolutional layer of a model. Additional visualizations of relative contribution are available in §I. Notably, in Fig. 6(a), an intriguing observation emerges: different input samples belonging to the same category display similar patterns in terms of soft relative contribution. This consistency suggests that channels making significant contributions to the classification of a particular category tend to exhibit consistency across diverse samples within that category. Essentially, specific categories are closely associated with fixed convolution filters and kernels.

Conversely, as depicted in Fig. 6(b), input samples from distinct categories manifest distinct patterns of soft relative contribution. This variability implies that channels contributing substantially to classification vary across different categories. In essence, each category is linked to a unique set of convolution filters and kernels.

Therefore, considering the $l$-th layer as an example, to identify components associated with a specific task (category), we initially select 1% of samples accurately classified for this category with the highest predicted probability. Subsequently, we average the features of these selected samples and calculate the relative contribution following the description in the main body of the paper.

# C    Relevant Feature Identifying with Backward Consideration

The computation of the relative contribution $\hat{r}_p^{(l)}$ for the input feature map $a_p^{(l)}$ in the $l$-th layer employs a backward approach. Consequently, it is imperative to account for the relative contribution of the input feature maps in the subsequent $(l+1)$-th layer when computing the relative contribution in the $l$-th layer. Specifically,

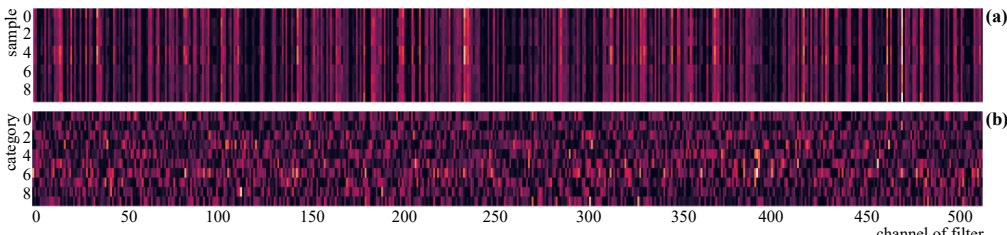

Figure 6: Visualization of the soft relative contribution $r_p^{(l)}$ (as defined in Eqn.(8)) in the 13-th convolutional layer of the VGG-16 model trained on the CIFAR-10 dataset. Subfigure (a) depicts different input samples belonging to the same category, while subfigure (b) showcases input samples from different categories. Bright and dark colors respectively represent large and small values of relative contribution.

the input feature maps in the $(l+1)$-th layer correspond to the output feature maps $\{a_q^{(l+1)}\}_{q=1}^{Q}$ of the $l$-th layer. Therefore, the relative contribution $\{\hat{r}_q^{(l+1)}\}_{q=1}^{Q}$ of these output feature maps in the $l$-th layer is equivalent to the relative contribution of the input feature maps in the $(l+1)$-th layer.

In accordance with Eqn.(2), if the relative contribution $\hat{r}_q^{(l+1)}$ for the output feature map $a_q^{(l+1)}$ is zero, then the relative contribution $\{r_{q,p}^{(l)}\}_{p=1}^{P}$ of the hidden feature maps $\{a_{q,p}^{(l)}\}_{p=1}^{P}$ is also considered as zero. Consequently, in Eqn.(9), the hard relative contribution $\hat{r}_{q,p}^{(l)}$ for the hidden feature map $a_{q,p}^{(l)}$ is recalculated as follows:

$$\hat{r}_{q,p}^{(l)} = \begin{cases} 1 & r_{q,p}^{(l)} \geq \alpha \text{ and } \hat{r}_q^{(l+1)} = 1 \\ 0 & r_{q,p}^{(l)} < \alpha \text{ or } \hat{r}_q^{(l+1)} = 0 \end{cases}. \tag{16}$$

The remaining equations in the main text remain unchanged.

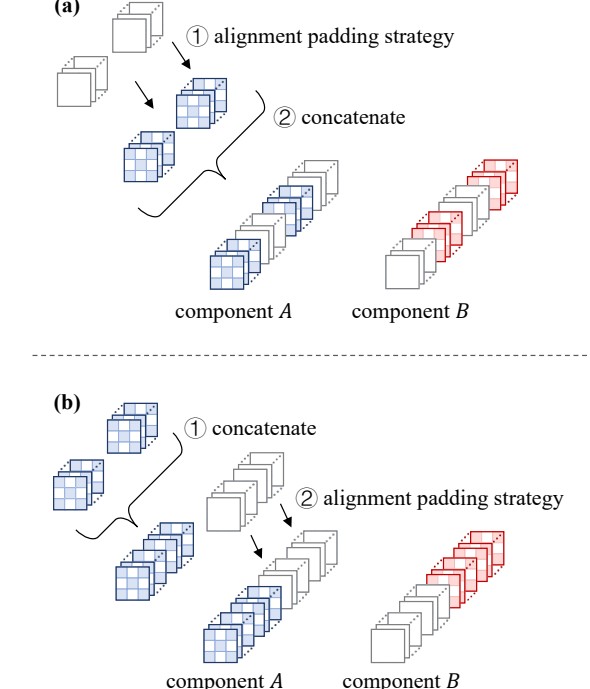

Figure 7: (a) The default alignment padding strategy and (b) the improved alignment padding strategy in the inception module.

# D Alignment Padding Strategy for Inception Module

Due to the distinctive multi-branch concatenation architecture, specifically the inception module, the default alignment padding strategy encounters challenges when applied to GoogleNet [23]. To address this, we have enhanced the alignment padding strategy specifically tailored for the inception module.

As illustrated in Fig. 7(a), the conventional alignment padding is employed in each convolutional filter by default, resulting in blank features calculated by zero kernels being dispersed throughout the channel at various locations after the concatenate operation. In our improved alignment padding strategy, depicted in Fig. 7(b), the alignment padding is applied after the concatenate operation.

# E More Experiments of CNN Model Disassembling

To further investigate the efficacy of the proposed disassembling method, we present the parameter size and Floating Point Operations Per Second (FLOPs) in Table 5. It is evident that both the parameter size (referred to as 'Para.') and FLOPs increase with the number of disassembled categories. For instance, the parameter size of the model disassembled for categories '3-9' from MNIST using VGG-16 is higher than that of the model disassembled for categories '0-2' from the same dataset and network.

Table 5: Comparison of disassembling performance using additional metrics. In 'Para.', 'Score1 / Score2' represent the parameter sizes (in millions, M) for the 'Disassembled Task' in the source and disassembled models, respectively. In 'FLOPs', 'Score1 / Score2' indicate the FLOPs (Floating Point Operations per Second) for the 'Disassembled Task' in the source model and the disassembled model, respectively.

| Dataset | Disassembled Task | VGG-16 | | ResNet50 | | GoogleNet | |
|---|---|---|---|---|---|---|---|
| | | Para. | FLOPs | Para. | FLOPs | Para. | FLOPs |
| MNIST | 0 | 33.65 / 4.59 | 33.29 / 12.31 | 23.52 / 13.77 | 130.47 / 105.06 | 6.31 / 4.39 | 55.43 / 46.40 |
| | 1 | 33.65 / 2.28 | 33.29 / 11.96 | 23.52 / 10.32 | 130.47 / 92.52 | 6.31 / 4.25 | 55.43 / 44.81 |
| | 0-2 | 33.65 / 9.62 | 33.29 / 26.65 | 23.52 / 18.39 | 130.47 / 122.13 | 6.31 / 5.30 | 55.43 / 49.04 |
| | 3-9 | 33.65 / 14.84 | 33.29 / 30.79 | 23.52 / 21.85 | 130.47 / 126.62 | 6.31 / 5.73 | 55.43 / 49.64 |
| FASHION-MNIST | 0 | 33.65 / 6.32 | 33.29 / 20.71 | 23.52 / 15.38 | 130.47 / 115.49 | 6.31 / 4.75 | 55.43 / 47.49 |
| | 1 | 33.65 / 3.17 | 33.29 / 13.66 | 23.52 / 12.18 | 130.47 / 94.90 | 6.31 / 4.53 | 55.43 / 45.91 |
| | 0-2 | 33.65 / 10.97 | 33.29 / 26.79 | 23.52 / 18.90 | 130.47 / 122.49 | 6.31 / 5.31 | 55.43 / 48.84 |
| | 3-9 | 33.65 / 15.27 | 33.29 / 30.30 | 23.52 / 21.78 | 130.47 / 127.69 | 6.31 / 5.74 | 55.43 / 49.12 |
| CIFAR-10 | 0 | 33.65 / 2.13 | 33.29 / 8.35 | 23.52 / 8.11 | 130.47 / 56.33 | 6.31 / 3.95 | 55.43 / 37.72 |
| | 1 | 33.65 / 2.35 | 33.29 / 11.85 | 23.52 / 8.41 | 130.47 / 77.19 | 6.31 / 4.21 | 55.43 / 40.99 |
| | 0-2 | 33.65 / 5.04 | 33.29 / 18.69 | 23.52 / 15.55 | 130.47 / 106.51 | 6.31 / 5.23 | 55.43 / 45.50 |
| | 3-9 | 33.65 / 13.48 | 33.29 / 27.36 | 23.52 / 20.70 | 130.47 / 116.74 | 6.31 / 5.73 | 55.43 / 46.74 |
| CIFAR-100 | 0 | 34.02 / 3.09 | 33.33 / 11.42 | 23.71 / 3.20 | 130.49 / 33.62 | 6.40 / 3.16 | 55.44 / 33.13 |
| | 1 | 34.02 / 3.11 | 33.33 / 12.73 | 23.71 / 6.18 | 130.49 / 55.76 | 6.40 / 3.22 | 55.44 / 35.43 |
| | 0-19 | 34.02 / 26.81 | 33.33 / 30.76 | 23.71 / 17.45 | 130.49 / 109.51 | 6.40 / 5.29 | 55.44 / 45.21 |
| | 20-69 | 34.02 / 30.01 | 33.33 / 31.97 | 23.71 / 21.56 | 130.49 / 120.01 | 6.40 / 5.57 | 55.44 / 47.08 |
| Tiny-ImageNet | 0 | 34.43 / 1.63 | 33.37 / 6.62 | 23.91 / 5.12 | 130.51 / 49.98 | 6.51 / 3.13 | 55.45 / 35.38 |
| | 1 | 34.43 / 1.58 | 33.37 / 4.66 | 23.91 / 4.41 | 130.51 / 38.59 | 6.51 / 3.13 | 55.45 / 34.21 |
| | 0-69 | 34.43 / 29.40 | 33.37 / 32.07 | 23.91 / 22.43 | 130.51 / 126.98 | 6.51 / 6.17 | 55.45 / 50.80 |
| | 70-179 | 34.43 / 30.72 | 33.37 / 32.61 | 23.91 / 23.23 | 130.51 / 128.33 | 6.51 / 6.23 | 55.45 / 51.13 |

Furthermore, our observations reveal that as the source model encompasses more categories, the individual components associated with each category tend to be smaller. For example, both the FLOPs and 'Para.' for the model disassembled component for category '0' from Tiny-ImageNet on VGG-16 are less than those for the same category '0' from MNIST on the same network. This suggests that the complexity and resource requirements of disassembled models are influenced not only by the number of categories they comprise but also by the inherent diversity and complexity of the datasets from which they are derived.

# F More Experiments of CNN Model Assembling

Table 6 provides detailed insights into the comparative performance of models with and without assembling. The results indicate that models with assembling generally exhibit reduced performance compared to those without assembling. For instance, the accuracy of '0-2 + 0-19' from CIFAR-10 + CIFAR-100 on VGG16 decreases after assembling. This decline in accuracy may be attributed to the possibility that samples from a specific category could inadvertently attain high confidence in other unrelated components of the assembled models. A notable example involves similar features across different categories, such as a monkey (from model $\mathcal{M}^{(1)}$) and a gorilla

Table 6: Performance comparison between models with and without assembling. '$\mathcal{M}^{(1)}$' and '$\mathcal{M}^{(2)}$' represent the accuracy of the two disassembled models in the 'Assembled Task', respectively. In 'Asse.', 'Score1 / Score2' indicate the average accuracy scores for the 'Assembled Task' in the assembled models, presented without fine-tuning and with ten epochs of fine-tuning, respectively. All results are expressed as percentages.

| Dataset | Assembled Task | VGG-16 | | | ResNet50 | | | GoogleNet | | |
|---|---|---|---|---|---|---|---|---|---|---|
| | | $\mathcal{M}^{(1)}$ | $\mathcal{M}^{(2)}$ | Asse. | $\mathcal{M}^{(1)}$ | $\mathcal{M}^{(2)}$ | Asse. | $\mathcal{M}^{(1)}$ | $\mathcal{M}^{(2)}$ | Asse. |
| MNIST + FASHION-MNIST | 0 + 0 | 100.00 | 100.00 | 97.35 / 97.41 | 100.00 | 100.00 | 62.05 / 87.32 | 100.00 | 100.00 | 96.05 / 96.12 |
| | 0-2 + 0-2 | 99.94 | 97.90 | 98.48 / 98.48 | 99.97 | 98.00 | 90.47 / 94.14 | 100.00 | 97.57 | 98.27 / 98.16 |
| | 0-2 + 3-9 | 99.94 | 95.57 | 95.62 / 96.19 | 99.97 | 95.66 | 88.58 / 93.20 | 100.00 | 96.44 | 96.34 / 96.52 |
| | 0-9 + 0-2 | 99.68 | 97.90 | 98.29 / 98.26 | 99.62 | 98.00 | 83.97 / 95.57 | 99.74 | 97.57 | 98.49 / 97.49 |
| CIFAR-10 + CIFAR-100 | 0 + 0 | 100.00 | 100.00 | 50.00 / 85.43 | 100.00 | 100.00 | 77.30 / 87.36 | 100.00 | 100.00 | 94.25 / 95.27 |
| | 0-2 + 0-19 | 95.47 | 82.50 | 74.17 / 74.19 | 97.43 | 77.15 | 64.34 / 76.37 | 98.17 | 87.55 | 79.22 / 79.22 |
| | 3-9 + 20-69 | 92.27 | 79.66 | 73.72 / 74.25 | 94.46 | 79.72 | 72.03 / 75.25 | 93.17 | 82.42 | 70.24 / 76.37 |
| | 0-9 + 20-99 | 91.37 | 72.54 | 72.07 / 73.18 | 92.32 | 76.39 | 65.97 / 74.65 | 92.38 | 78.04 | 66.59 / 70.36 |
| CIFAR-10 + Tiny-ImageNet | 0 + 0 | 100.00 | 100.00 | 94.70 / 90.72 | 100.00 | 100.00 | 86.95 / 94.51 | 100.00 | 100.00 | 62.40 / 80.34 |
| | 0-2 + 0-69 | 99.94 | 55.49 | 53.20 / 53.20 | 97.43 | 56.40 | 43.09 / 56.38 | 98.17 | 59.40 | 57.51 / 57.81 |
| | 3-9 + 0-69 | 92.27 | 55.49 | 50.20 / 52.48 | 94.46 | 56.40 | 52.14 / 58.63 | 93.17 | 59.40 | 53.74 / 55.32 |
| | 0-9 + 70-179 | 91.37 | 47.95 | 42.30 / 47.98 | 92.32 | 53.42 | 47.21 / 55.17 | 92.38 | 51.04 | 48.00 / 52.16 |
| CIFAR-100 + Tiny-ImageNet | 0 + 0 | 100.00 | 100.00 | 50.00 / 76.28 | 100.00 | 100.00 | 53.00 / 87.34 | 100.00 | 100.00 | 50.00 / 85.28 |
| | 0-19 + 0-69 | 82.50 | 55.49 | 50.66 / 57.19 | 77.15 | 56.40 | 57.86 / 69.23 | 87.55 | 59.40 | 58.67 / 69.14 |
| | 20-69 + 70-179 | 79.66 | 47.95 | 50.08 / 65.71 | 79.72 | 53.42 | 56.53 / 69.79 | 82.42 | 51.04 | 54.09 / 71.27 |
| | 0-99 + 0-199 | 71.38 | 47.41 | 43.06 / 56.13 | 73.97 | 53.51 | 53.05 / 57.23 | 76.08 | 50.53 | 48.66 / 58.28 |

Table 7: The disassembling performance on ImageNet with VGG-16. 'Base.' denotes the average accuracy for specific categories from the source model. For 'Disa.', 'Score1 (+*Score2*)' denotes the average accuracy 'Score1' (the improved average accuracy '*Score2*' compared to 'Base.') for specific categories from the disassembled model.

| Dataset | Disassembled Task | VGG-16 | |
|---|---|---|---|
| | | Base. (%) | Disa. (%) |
| ImageNet | 0 | 90.00 | 100.00 (+*10.00*) |
| | 1 | 90.00 | 100.00 (+*10.00*) |
| | 0-9 | 81.20 | 83.37 (+*2.17*) |
| | 0-99 | 76.06 | 79.24 (+*3.18*) |
| | 0-299 | 75.84 | 77.23 (+*1.39*) |
| | 100-499 | 73.15 | 74.38 (+*1.23*) |
| | 500-999 | 66.31 | 67.34 (+*1.03*) |

(from model $\mathcal{M}^{(2)}$). In such cases, a sample from one category might receive a higher confidence score from the model trained on the other category.

However, a key advantage of model assembling lies in the reduction of inference time. As the number of disassembled components increases, running each component independently becomes more time-consuming compared to utilizing an assembled model. This underscores the trade-off between model performance and computational efficiency in the assembling process.

## G  More Experiments of CNN Model on Large-scale Datasets

It is crucial to investigate the scalability of the proposed framework on large-scale datasets, such as ImageNet [24]. Table 7 presents the disassembling performance on ImageNet with VGG-16, demonstrating that the performance of the disassembled model surpasses that of the source model in all cases. Specifically, when disassembling categories '0-99', the accuracy of the disassembled model is higher by 3.18% compared to the source model. These experiments underscore the scalability of the proposed method, which extends its applicability to mainstream benchmark datasets.

## H  MDA Applied to Other Domains

The proposed MDA method demonstrates significant flexibility and utility beyond its primary application. Specifically, it enables the customization and reuse of pre-trained CNN classifiers for specific tasks without

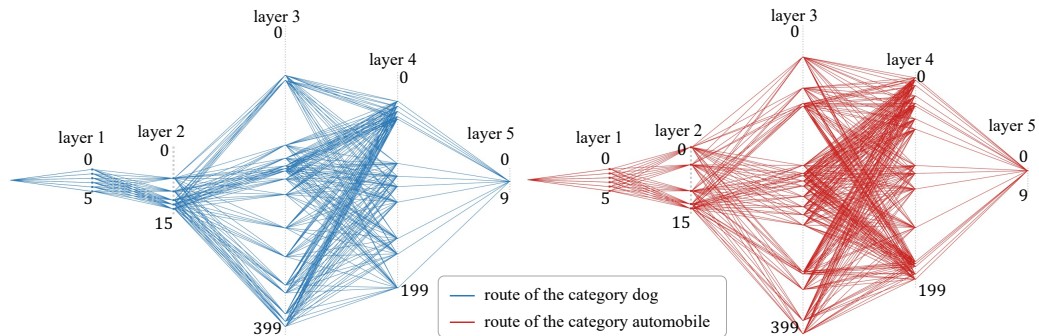

Figure 8: Decision routes of the categories 'dog' and 'automobile' in the LeNet-5 model on the CIFAR-10 dataset, where the channels of the 'dog' and 'automobile' in layer 1 are the same, while in later layers, such as layer 2, 3, 4, and 5, they are totally different.

Table 8: Comparison of model compression performance. 'Base.' refers to the source model, while 'Ours.' represents the model compressed using the proposed method. 'Acc.', 'FLOPs', and 'Para.' indicate the accuracy, floating point operations, and parameter size of the model, respectively. All accuracy scores are presented as percentages.

| Dataset | CNN classifier | Base. | | | Ours | | | FPGM | | | HRank | | |
|---|---|---|---|---|---|---|---|---|---|---|---|---|---|
| | | Acc. | FLOPs | Para. | Acc. | FLOPs | Para. | Acc. | FLOPs | Para. | Acc. | FLOPs | Para. |
| CIFAR-10 | VGG-16 | 92.90 | 33.29 | 33.65M | 91.31 | 28.00 | 18.63M | 91.58 | 28.87 | 26.23M | 91.97 | 20.19 | 13.25M |
| | ResNet50 | 94.15 | 130.47 | 23.52M | 93.71 | 123.21 | 18.93M | 93.24 | 63.71 | 12.28M | 93.89 | 71.12 | 14.42M |
| | GoogleNet | 93.76 | 55.43 | 6.31M | 92.38 | 47.25 | 5.78M | 93.22 | 25.21 | 3.93M | 93.13 | 27.16 | 4.09M |

necessitating additional training. This adaptability extends to various other tasks, including but not limited to model decision route analysis, model compression, knowledge distillation.

## H.1 Model Decision Route Visualization

The concept of a 'decision route' refers to the specific data flow pathways utilized for each category within a deep learning model. These pathways are typically sparse and remain fixed. As elucidated in §B, the prediction for different categories within a model hinges on the large contributions from distinct feature maps, meaning each category is associated with its own set of relevant model parameters. The proposed framework for contribution allocating and aggregating, combined with component locating techniques, facilitates the construction of unique decision routes for each category.

Such decision route analysis offers several benefits. Firstly, it enhances the explainability of deep models. By comparing the decision route of a misclassified sample with the corresponding correct category's route, researchers can identify where the data flow diverged incorrectly. Additionally, these visualizations serve as powerful tools for a deeper understanding and exploration of deep classifiers.

For example, Fig.8 illustrates decision routes for specific categories using LeNet-5 [55], which includes two convolutional layers and three linear layers. It is observable that each category possesses distinct decision routes, yet there is some overlap among them. This overlap can be attributed to the fact that shallower convolutional filters often process fundamental features like color, texture, and edges—aligning with existing research [8, 9, 10, 12, 13, 14] and the mechanisms of biological visual information processing [5, 6, 7]. In the deeper layers, particularly the final linear layer, we notice that connections between neurons are sparse yet fixed, reflecting the emergence of category-specific features at deeper levels of the CNN.

In summary, visualizing the decision routes for specific classes within a CNN classifier not only aids in dissecting the classifier's underlying mechanisms but also proves instrumental in debugging and enhancing the model's performance.

## H.2 Model Compression

The proposed MDA presents a novel approach to model compression for CNN classifiers. Utilizing our component locating technique, we effectively disassemble category-specific parameters from the source model. This process entails separating parameters utilized across all categories and discarding those that are redundant, i.e., not used by any category. Consequently, this method offers a unique avenue for model compression.

Table 9: Comparison of knowledge distillation performance. 'Base.' represents the average accuracy for the 'Assembled Task' obtained from the source models. All accuracy scores are expressed as percentages.

| Dataset | Distilled Category | VGG-16 Base. | Ours | KA |
|---|---|---|---|---|
| MNIST + FASHION-MNIST | 0 + 0 | 94.60 | 97.35 | 92.24 |
| | 0-2 + 0-2 | 96.69 | 98.48 | 93.81 |
| | 0-2 + 3-9 | 96.12 | 95.62 | 92.76 |
| | 0-9 + 0-2 | 98.24 | 98.29 | 93.57 |
| CIFAR-10 + CIFAR-100 | 0 + 0 | 89.20 | 94.00 | 84.35 |
| | 0-2 + 0-19 | 74.03 | 74.17 | 69.46 |
| | 3-9 + 20-69 | 75.16 | 73.72 | 71.38 |
| | 0-9 + 20-99 | 74.87 | 72.07 | 71.60 |
| CIFAR-10 + Tiny-ImageNet | 0 + 0 | 88.20 | 94.70 | 85.62 |
| | 0-2 + 0-69 | 51.97 | 53.20 | 52.67 |
| | 3-9 + 0-69 | 54.02 | 50.20 | 53.26 |
| | 0-9 + 70-179 | 49.33 | 42.30 | 51.84 |
| CIFAR-100 + Tiny-ImageNet | 0 + 0 | 83.00 | 85.50 | 81.31 |
| | 0-19 + 0-69 | 69.86 | 50.66 | 68.24 |
| | 20-69 + 70-179 | 71.48 | 50.08 | 67.52 |
| | 0-99 + 0-199 | 55.97 | 43.06 | 52.74 |

For empirical validation, we conducted compression experiments using three mainstream classifiers: VGG-16 [56], ResNet-50 [22], and GoogleNet [23] on the CIFAR-10 dataset [19], as detailed in Table 8. Additionally, we benchmarked our method against state-of-the-art (SOTA) model compression techniques, specifically two pruning-based methods: FPGM [57] and HRank [58].

The results, as shown in Table 8, indicate that our method achieves comparable accuracy to the SOTA methods, FPGM and HRank. However, it is observed that our method exhibits higher Floating Point Operations Per Second (FLOPs) and parameter size compared to these pruning-based methods. This distinction likely stems from our method's focus on disassembling parameters relevant to each category, as opposed to pruning methods which aim to filter out parameters irrelevant to all categories. Consequently, while our method retains parameters commonly used across categories, pruning methods like FPGM may discard some of these components without significantly affecting the final prediction, leading to fewer FLOPs and a lower parameter size.

In future research, we aim to delve deeper into the capabilities and potential of our disassembling and assembling approach in the realm of model compression, exploring ways to enhance its efficiency and effectiveness in reducing model complexity while preserving or even enhancing performance.

## H.3 Knowledge Distillation

The proposed MDA focuses on assembling task-aware components disassembled from different models into a new, unified model. This process bears resemblance to Knowledge Amalgamating (KA) as described in Shen et al. [59], where knowledge from multiple 'teacher' models is distilled into a single 'student' model. While the overarching goals of MDA and KA are similar, the techniques employed in each approach differ significantly.

To assess the efficacy of our MDA method in the context of knowledge distillation, we conducted comparative experiments against KA using the VGG-16 [21] model on five benchmark datasets: MNIST [55], Fashion-MNIST [60], CIFAR-10 [19], CIFAR-100 [19], and Tiny-ImageNet [20]. The results of these experiments are summarized in Table 9.

The results indicate that the proposed MDA method generally achieves higher accuracy than KA, particularly in scenarios with a smaller number of assembled categories, such as the combination of 'MNIST + FASHION-MNIST'. Conversely, as the number of assembled categories increases—for instance, in the case of 'CIFAR-100 + Tiny-ImageNet', the performance of our method tends to decline, even falling below that of the source model and KA. This decrease in accuracy could be attributed to the increased complexity and potential interference when assembling a larger number of task-aware model components. Specifically, for a test image from an unknown category, the flow through all decision routes in the assembled model can lead to confusion and incorrect predictions, particularly if the components are primarily tailored for categories in a different dataset (e.g., dataset 'A'), thus adversely affecting the accuracy of predictions for samples in dataset 'B'.

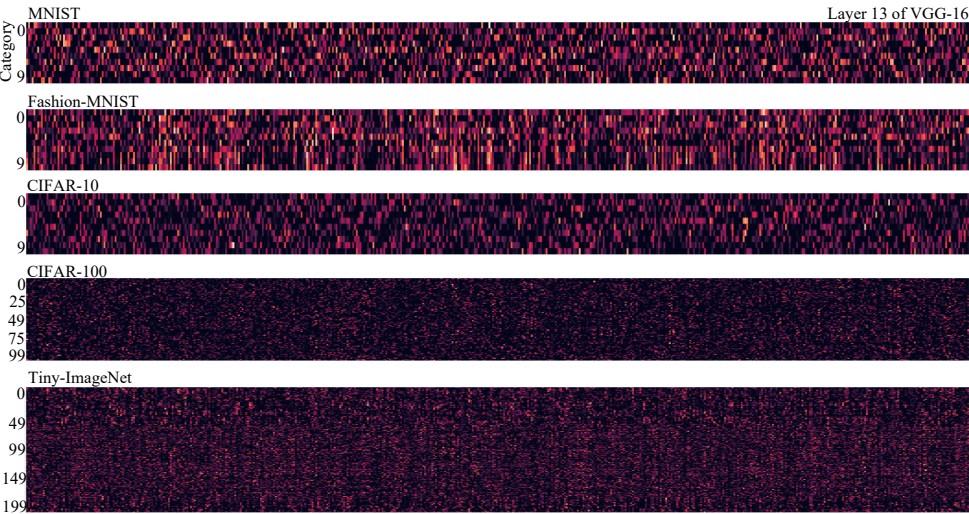

Figure 9: Visualization of the soft relative contribution $r_p^{(l)}$ (as defined in Eqn.(8)) for input samples from different categories in layer 13 of VGG-16 on different datasets.

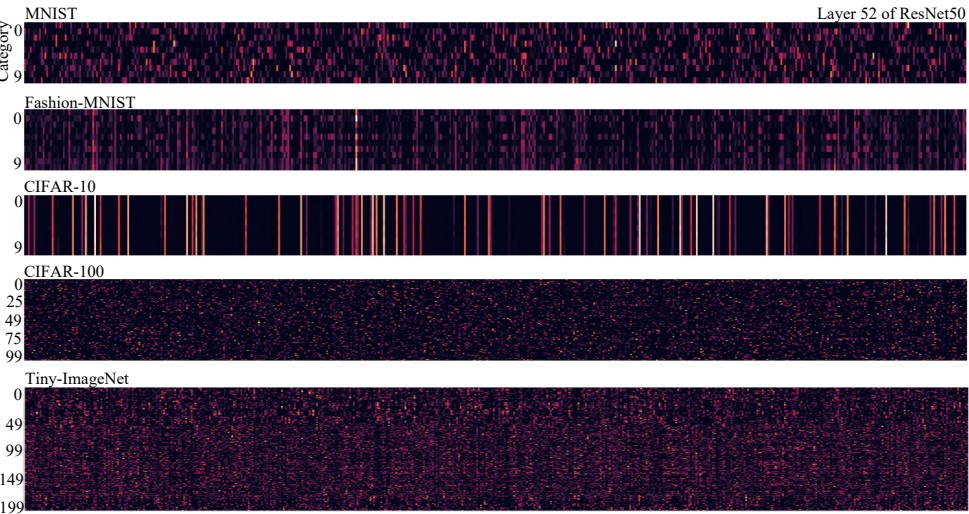

Figure 10: Visualization of the soft relative contribution $r_p^{(l)}$ (as defined in Eqn.(8)) for input samples from different categories in layer 52 of ResNet-50 on different datasets.

# I  More Visualization of Relative Contribution (§B)

In this section, Figs. 9-12 show additional visualization results of the relative contribution. As depicted in Figs. 9-11, for a specific layer of the networks on different datasets, different categories exhibit distinct patterns of the soft relative contribution. From Fig. 12, it is evident that in different layers of VGG-16, different categories demonstrate varied patterns of the soft relative contribution. These visualizations offer additional insights into the associations between categories and specific filters across various layers of the neural networks. In certain instances, such as in the case of Layer 52 of ResNet50 on CIFAR-10, distinct categories exhibit comparable substantial relative contributions across diverse channels. The potential explanation for this phenomenon lies in the insufficient formation of category-related features within this layer, attributable to the limited number of categories in CIFAR-10 (only ten), coupled with the residual structure and depth of the network. Nonetheless, other layers manifest disparate substantial relative contributions for distinct categories across diverse channels. This phenomenon stands as a pivotal factor in ensuring the accurate prediction performance of ResNet50 on the CIFAR-10 dataset.

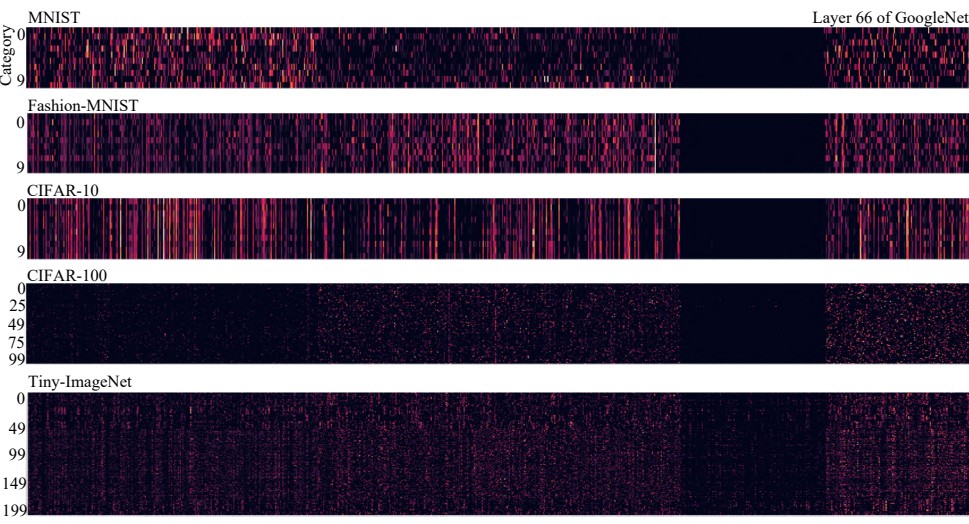

Figure 11: Visualization of the soft relative contribution $r_p^{(l)}$ (as defined in Eqn.(8)) for input samples from different categories in layer 66 of GoogleNet on different datasets.

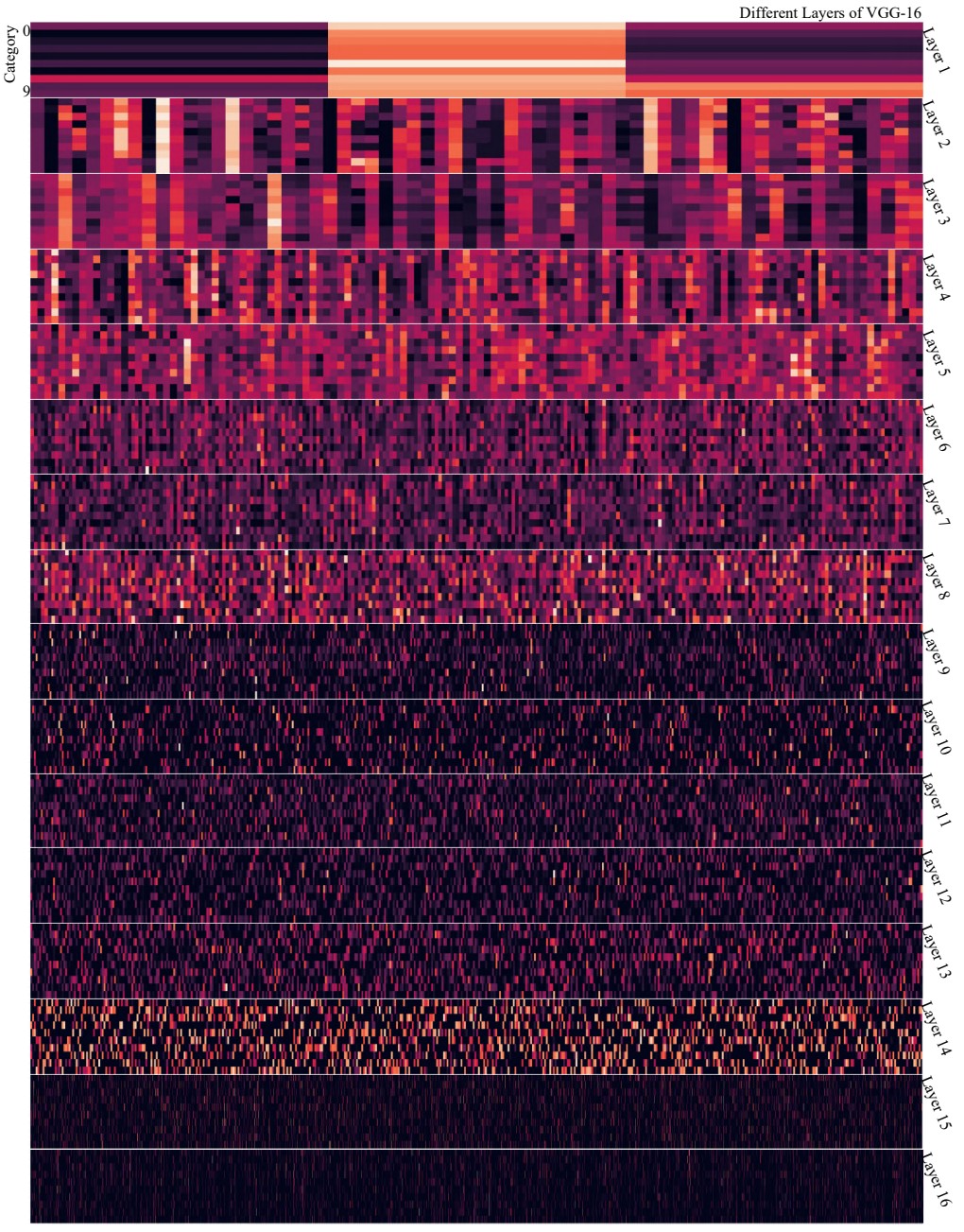

Figure 12: Visualization of the soft relative contribution $r_p^{(l)}$ (as defined in Eqn.(8)) for input samples from different categories in different layers of VGG-16 on CIFAR-10.

