# OpenReview forum: "Model LEGO: Creating Models Like Disassembling and Assembling Building Blocks"
_NeurIPS.cc/2024/Conference — NeurIPS 2024 poster_

### Official Review · Reviewer_17d9 · 2024-07-02

**Soundness:** 4
**Presentation:** 3
**Contribution:** 3
**Rating:** 7
**Confidence:** 5

**Summary:**

This paper proposes a novel framework for model disassembling and assembling. A component locating technique is introduced to disassemble task-aware components from the models. And an alignment padding strategy and a parameter scaling strategy are also designed to assemble these useful components.

This work is meaningful to the interpretability of deep neural network models, as it combines the theory of biology and brain science theory to design human-interpretable models, which can uncover the black box CNN.

**Strengths:**

The paper is clearly written and easy to follow.
The proposed task is very novel, and the whole pipeline is well presented and easy to understand, and makes sense.
The locating-then-assembling paradigm is well supported by the informative components identification and alignment padding strategy.
The authors tested their proposed method on various popular CNN models and obtained convincing results.

**Weaknesses:**

In section 3.1.1, the insight and rationale of the proposed metric is not clear. Why can this metric measure the contribution of features?

Also, in the 4.2, why the Parameter Scaling Strategy can balance the effects of different componens?

**Questions:**

refer to the weakness part.

**Limitations:**

This paper only design CNN-based method, ignoring the popular Vit model.

---

> ### Author Rebuttal · Authors · 2024-08-06
>
> Thank you for your review and comments. We are pleased that you find our work to be novel and that you appreciate the clarity of our proposed pipeline and the convincing nature of the results. Below are our responses to each of your comments (your comments are highlighted in italics).
>
> > Q1: *In section 3.1.1, the insight and rationale of the proposed metric is not clear. Why can this metric measure the contribution of features?*
> >
>
> We apologize for the confusion. The "contribution of features" refers to the extent to which features impact the prediction results (as described in lines 97 to 99 of the paper). This contribution is dynamic and continuous throughout the network layers during inference (lines 100 to 104). Based on this concept, we introduced two mechanisms for contribution flow: contribution aggregation and contribution allocation.
>
> It is important to note that contributions are relative, meaning that a larger feature value has a greater relative impact on the model's final prediction (lines 129 to 131). In Sections 3.1.1 and 3.1.2, the contribution of features during aggregation and allocation is defined by their relative magnitude (Equations 5 and 8). Additionally, to account for nonlinear functions such as ReLU, we only use features with positive values to compute this contribution (Equations 4 and 7). Our experiments demonstrate that this method of calculating contributions effectively identifies key parameters related to subtasks.
>
> > Q2: *Also, in the 4.2, why the Parameter Scaling Strategy can balance the effects of different components?*
> >
>
> We apologize for any confusion. The term "effects of different components" refers to the varying magnitudes of parameters from different source models, which can influence the output disproportionately. Even if an input does not belong to a particular task-aware component, if the parameters of this component are of a significantly larger magnitude, it can produce a disproportionately large response, thereby skewing the prediction (lines 228 to 232 of the paper).
>
> The Parameter Scaling Strategy addresses this issue by scaling the parameters of each task-aware component before assembly. This scaling is based on the response of each component to the same input, ensuring that the magnitudes of the parameters are kept at a consistent level. This prevents any single component from disproportionately affecting the prediction. Ablation experiments presented on lines 316 to 320 of the paper demonstrate the effectiveness of this strategy. We hope this explanation clarifies your concern.
>
> > Q3: Limitations: This paper only designs CNN-based methods, ignoring the popular ViT model.
> >
>
> We understand your concern.  The proposed MDA method, including the underlying concepts of contribution aggregation and allocation, is indeed applicable to Transformer models. The method is not tightly coupled with CNNs and can be applied to linear layers and other structures beyond convolutional layers (Appendix C), making it a general paradigm for model disassembly and reassembly tasks. This is demonstrated in the experiments presented in Section 5.3 of the paper. Additionally, we conducted a preliminary model disassembly experiment using ViT-Tiny on ImageNet-10, and the results are shown in Table 1.
>
> Table 1: Disassembly performance of ViT-Tiny on ImageNet-10.
>
> | Disassembled Task | Base (%) | Disa (%) |
> | --- | --- | --- |
> | 0-1 | 87.70 | 89.25 (+1.55) |
> | 1-3 | 75.10 | 78.10 (+3.00) |
> | 3-9 | 77.51 | 79.29 (+1.78) |
>
> As shown in Table 1, the disassembled model can still be used for inference, with accuracy exceeding that of the original model.
>
> Additionally, we further tested the assembly performance of ViT-Tiny on CIFAR-10 and CIFAR-100. However, the results were not satisfactory. For instance, when assembling disassembled CIFAR-10(0-3) and CIFAR-100(11-17) models, the accuracy without fine-tuning was only 13.27%. The primary reason for this is the task interference that occurs when submodels from different source models are assembled, a phenomenon that is particularly severe due to the self-attention mechanism inherent in Transformers. We believe that adjusting parameter distribution from different submodel parameter spaces before assembly could be a potential solution to this issue, which will be a focus of our future work.

---

> > ### Comment · Reviewer_17d9 · 2024-08-11
> > **Good response**
> >
> > This response successfully addressed my concerns. I maintain the accept score

---

> > > ### Author Response · Authors · 2024-08-13
> > >
> > > We are pleased that our responses have addressed your concerns. Thank you for your constructive feedback and support for our work.

---

### Official Review · Reviewer_gfoc · 2024-07-03

**Soundness:** 4
**Presentation:** 3
**Contribution:** 4
**Rating:** 7
**Confidence:** 5

**Summary:**

This paper draws inspiration from the information subsystem pathways in biological vision systems and proposes a Model Disassembling and Assembling (MDA) approach.

- For model disassembling, the authors introduce the concepts of contribution aggregation and contribution allocation within convolutional filters and their kernels. The relative contribution of features is calculated and linked to the corresponding parameters for subsequent structure pruning.

- For model assembling, a new model is established by combining the disassembled, task-aware components derived from different source models without necessitating retraining.

To validate the effectiveness of MDA, the authors conduct extensive experimental evaluations across diverse architectures, including CNNs and GCNs. The results show that MDA can work effectively on these architectures. In addition to model creation, the authors also explore potential applications of MDA, such as model decision route analysis and model compression.

**Strengths:**

1. The concept of Model Disassembling and Assembling (MDA) is novel, offering a new perspective on model architecture. It suggests that each part of a model related to specific tasks or categories can be considered an independent functional component. These components can be flexibly reused and assembled to achieve various combinations for multi-classification tasks.

2. The concept of MDA, along with the proposed methods and formula definitions, is clearly presented. The experiments are well designed, and the comprehensive results validate that MDA can work effectively for model reuse or creation, and other task scenarios. Moreover, the extensive experiments provide deeper insights into MDA, enhancing the overall quality of the paper.

3. The paper is written in a clear and accessible manner. The illustrations, methods, and formulas are easy to understand, and the language used is straightforward. The provided source codes are also well-structured and easy to read.

4. The proposed method demonstrates significant performance benefits. Besides creating models by reusing model components, the basic idea of MDA has the potential to inspire future research directions, such as model interpretability and model architecture design.

**Weaknesses:**

1. As the number of categories increases, the performance of disassembling decreases. This means that to achieve the best disassembling effect, it is necessary to decompose each category separately, which is time-consuming and may result in a heavier assembled model.

2. Manually setting thresholds (Eqns. 9 and 10) to obtain relative contributions may not guarantee consistently good results across tasks.

3. While the authors claim that model assembling does not necessitate retraining or incur performance loss (Line 204), the assembled models are often inferior to the baseline without retraining or fine-tuning (see Table 1).

4. In the assembled model, different categories fail to follow their own pathways extracted from the model disassembling. For example, a disassembled conv-filter of CIFAR-10-cat0 might generate a high response on CIFAR-100-cat0 because the conv-filter has never seen CIFAR-100-cat0, which can be regarded as inter-class interference. This is why the assembled model initially shows degraded performance. Ideally, model assembling should ensure that pathways between different categories are mutually exclusive.

**Questions:**

1. The results in Table 3 have shown the effectiveness of MDA when applied to GCN models. However, could the authors provide more details on the design of MDA for GCN models?

2. Does model disassembling require additional training? If so, please present the details on how the numerical results in Tables 1 and 3 were obtained.

3. The fine-tuning of the assembled model will inevitably cause a shift in the internal pattern of the model. Do the pathways of the fine-tuned model remain the same as the original ones?

4. Have the authors considered the interference between categories when assembling the disassembled components? Are there any potential solutions to this issue?

**Limitations:**

The limitations have been discussed in the paper. Additionally, there is no societal impact from the work performed.

---

> ### Author Rebuttal · Authors · 2024-08-06
>
> Thank you for your review and comments. We are pleased that you find our work on model disassembly and reassembly to be novel and consider it to be inspiring for model interpretability and architectural design. Below are our responses to each of your comments (your comments are highlighted in italics).
>
> > Q1: As the number of categories increases, the performance of disassembling decreases. This means that to achieve the best disassembling effect, it is necessary to decompose each category separately, which is time-consuming and may result in a heavier assembled model.
> >
>
>  Thank you for your comment. We respectfully disagree with this viewpoint. Firstly, there is no evidence suggesting that the performance of disassembling decreases with an increasing number of categories. In fact, as shown in Table 1, regardless of the number of categories into which the model is disassembled, the performance of the disassembled models generally exceeds that of the original model, though the degree of improvement may vary. This is because model disassembly extracts only the parameters relevant to the target categories, reducing interference from unrelated parameters and thereby enhancing prediction accuracy. More importantly, the time required for model disassembly is minimal compared to training a new model, as detailed in our response to the first comment by Reviewer FhUd.
>
> > Q2: *Manually setting thresholds (Eqns. 9 and 10) to obtain relative contributions may not guarantee consistently good results across tasks.*
> >
>
> Thank you for your feedback. We agree that manually setting thresholds (Equations 9 and 10) may not consistently yield optimal results across different tasks. The choice of thresholds represents a trade-off between the performance of the disassembled model and the number of model parameters. Consequently, the optimal thresholds for Equations 9 and 10 may vary across different tasks. To address the impact of manually chosen thresholds on the results, we are currently exploring a more automated approach. This involves developing adaptive algorithms that dynamically adjust the thresholds to meet the specific requirements of different tasks. Thank you again for your valuable suggestion.
>
> > Q3: *While the authors claim that model assembling does not necessitate retraining or incur performance loss (Line 204), the assembled models are often inferior to the baseline without retraining or fine-tuning (see Table 1).*
> >
>
> Thank you for your comment. The table you are referring to should be Table 2. We acknowledge that in some scenarios, the performance of assembled models without fine-tuning can be inferior to that of the original models. As discussed on lines 270-272 of the paper, the assembled submodels originate from different source models, and even the data used to train these original models may differ. Consequently, during prediction, the numerous parameters of these submodels can interfere with each other, affecting the accuracy of the assembled model. This is indeed a limitation of our current approach and an area for future work, as mentioned on lines 341-343 of the paper.
>
> However, this method still holds significant value. Currently, to address this limitation, we fine-tune the assembled models to mitigate interference among the submodels. It is important to note that this fine-tuning requires only a few iterations (e.g., the fine-tuned results in Table 2 are based on 10 iterations), which is substantially less computationally expensive than training a model from scratch. In some cases, the performance of the final fine-tuned model even exceeds that of the original model. Notably, fine-tuning is not required during the model disassembly phase.
>
> > Q4: *In the assembled model, different categories fail to follow their own pathways extracted from the model disassembling. For example, a disassembled conv-filter of CIFAR-10-cat0 might generate a high response on CIFAR-100-cat0 because the conv-filter has never seen CIFAR-100-cat0, which can be regarded as inter-class interference. This is why the assembled model initially shows degraded performance. Ideally, model assembling should ensure that pathways between different categories are mutually exclusive.*
> >
>
> Thank you for your detailed comment. We understand your concerns and agree with your observations. Indeed, the failure of certain categories to adhere to their own pathways extracted from the model disassembly can result in degraded performance due to what you described as "inter-class interference." To address this issue, we have implemented the following measures during model assembly:
>
> 1. Parameter Scaling Strategy: During model assembly, we apply a parameter scaling strategy to mitigate significant differences in parameter magnitudes between disassembled submodels. This helps prevent the assembled model's predictions from being disproportionately influenced by submodels with larger parameter magnitudes. The results of our ablation studies on this strategy are presented in Table 4.
> 2. Minor Fine-Tuning Mechanism: As mentioned in our response to your previous comment, we consider performing a small number of fine-tuning iterations after model assembly. This helps the model adapt to the new data distribution and can enhance overall performance.
>
> Additionally, we are exploring more generalized and stable methods for assembling submodels to prevent the occurrence of "inter-class interference." Thank you once again for your valuable feedback.

---

> ### Author Response · Authors · 2024-08-06
> **Rebuttal by Authors [Q5-Q8]**
>
> > Q5: *The results in Table 3 have shown the effectiveness of MDA when applied to GCN models. However, could the authors provide more details on the design of MDA for GCN models?*
> >
>
> We apologize for any confusion. We are pleased that Table 3 demonstrates the effectiveness of MDA on GCN models. In fact, the core computation of GCN models is given by $\mathbb{H}^{l+1} = \delta(\tilde{\mathbb{A}}\mathbb{H}^{l}\mathbb{W}^{l})$, where $\tilde{\mathbb{A}}$ is the degree-normalized adjacency matrix, $\mathbb{H}^{l}$ and $\mathbb{W}^{l}$ represent the features and weights at layer $l$, respectively, and $\delta$ denotes an activation function.
>
> This process can be simplified to linear operations involving $\mathbb{H}^{l}\mathbb{W}^{l}$. The MDA method proposed in our paper, which is designed for convolutional operations (discussed in Sections 3 and 4 of the paper), is also applicable to linear operations (as detailed in Appendix C). Therefore, no special modifications are required to apply MDA to GCN models; the same MDA method used for CNNs can be directly employed. This underscores both the versatility and effectiveness of the MDA method.
>
> > Q6: *Does model disassembling require additional training? If so, please present the details on how the numerical results in Tables 1 and 3 were obtained.*
> >
>
> Thank you for your question. Model disassembling does not require any additional training. The performance results presented in Tables 1 and 3 are derived directly from the disassembled models. As observed, the performance of these disassembled models generally exceeds that of the original models. The detailed reasons for this performance improvement can be found in our response to your first comment.
>
> > Q7: *The fine-tuning of the assembled model will inevitably cause a shift in the internal pattern of the model. Do the pathways of the fine-tuned model remain the same as the original ones?*
> >
>
> This is an interesting question. We have disassembled both the pre-fine-tuned and fine-tuned assembled models to examine the internal decision pathways. Our analysis reveals that fine-tuning does indeed result in changes to the internal decision pathways of the model. These changes allow the fine-tuned model to better adapt to the new data distribution, thereby mitigating the interference between parameters from different categories during inference.
>
> > Q8: *Have the authors considered the interference between categories when assembling the disassembled components? Are there any potential solutions to this issue?*
> >
>
> Thank you for your question. We have indeed considered the issue of category interference when assembling disassembled components. Details regarding this concern are addressed in our responses to your third and fourth comments.
>
> In the paper, we propose two approaches to mitigate this interference: the "parameter scaling strategy" and "minimal fine-tuning mechanism." These methods can alleviate inter-category interference to some extent without incurring significant computational overhead. Furthermore, in our future work, we plan to explore adaptive parameter adjustment strategies from the perspective of the parameter space of submodels. For example, we are considering leveraging diffusion models to adjust parameter distributions. This approach would involve adapting the parameters of the submodels based on the characteristics of each category, enabling the assembled model to rapidly adapt to new data distributions and potentially eliminating the need for fine-tuning.
>
> However, it is important to note that the current methods for model disassembly and assembly still hold significant value.

---

> > ### Comment · Reviewer_gfoc · 2024-08-12
> >
> > Thank you for your detailed response, which effectively addressed my concerns.  I have also reviewed the comments and responses provided to the other reviewers, and I find that they align with my understanding of this work.
> >
> > I believe that the model disassembly and assembly presented in this work are both insightful and significant for the deep learning community. I also look forward to seeing its potential application to a broader range of models and tasks.
> >
> > Based on the above considerations, I will maintain a positive recommendation for the acceptance of this work.

---

> > > ### Author Response · Authors · 2024-08-13
> > >
> > > We are very pleased that these responses address your concerns. Thank you for your valuable feedback and recognition of our work.

---

### Official Review · Reviewer_UTXG · 2024-07-08

**Soundness:** 2
**Presentation:** 2
**Contribution:** 2
**Rating:** 2
**Confidence:** 3

**Summary:**

This paper proposes the Model Disassembling and Assembling (MDA) task for CNN classifiers, introducing techniques for extracting and reassembling task-aware components. Experiments show reassembled models perform comparably or better than original models. The approach offers new applications in decision route analysis, model compression, and knowledge distillation, with future extensions planned for Transformers and multi-modal models.

**Strengths:**

- The idea presented by this paper is novel and the technique adopted for solving the problem is new.

**Weaknesses:**

1. The definition of sub-task is strongly tied to category, severely limiting it to classification tasks. Additionally, the entire method seems complicated, appearing to be overfitted to classification tasks and the CNN architecture. Its inability to handle other types of tasks, especially self-supervised learning, which is widely recognized as the future, raises concerns about the value of this work.
2. There is a lack of systematic comparisons with other works, such as deep model reassembly[1], and the study only includes self-constructed specific tasks, making it hard to evaluate the technical soundness.
3. Given the widespread adoption of transformers in the field, the absence of experiments involving transformers is unfortunate.

[1] Yang, Xingyi, et al. "Deep model reassembly." Advances in neural information processing systems 35 (2022): 25739-25753.

**Questions:**

Could you provide an intuitive explanation for why your method sometimes outperforms the baseline, given that the baseline is end-to-end trained for the classification task?

**Limitations:**

As stated in the weakness part, the absence of the results on widely-adopted transformer architectures and its inapplicability to models trained with tasks other than classification (e.g., masked image modeling) may limit the value of this work.

---

> ### Author Rebuttal · Authors · 2024-08-06
>
> Thank you for your review and comments. We are pleased that you found our work to be novel and our techniques for solving the problem to be innovative. Below are our responses to each of your comments (your comments are highlighted in italics).
>
> > Q1: *The definition of sub-task is strongly tied to category, severely limiting it to classification tasks. Additionally, the entire method seems complicated, appearing to be overfitted to classification tasks and the CNN architecture. Its inability to handle other types of tasks, especially self-supervised learning, which is widely recognized as the future, raises concerns about the value of this work.*
> >
>
> Thank you for your constructive comments. We highly value the points you raised and provide our responses below.
>
> **On the Definition of Sub-tasks**: The definition of sub-tasks in our model disassembly and reassembly is not tightly bound to categories. Beyond classification tasks, sub-tasks can be defined in various ways, including:
>
> 1. Image-related Sub-tasks: Besides classification, sub-tasks can extend to other image-related tasks, such as object detection and image segmentation. For instance, sub-tasks could include detecting/segmenting "pedestrians" or "vehicles." In image generation, sub-tasks could involve generating "eyes" or "noses." Therefore, the concept of model disassembly and reassembly is not strictly limited to classification tasks but has potential applications in detection, generation, and more.
> 2. Natural Language-related Sub-tasks: Additionally, sub-tasks can be defined for natural language processing tasks. In tasks like natural language generation or question answering, factual knowledge such as "The Eiffel Tower is located in Paris" can be defined as a sub-task. Queries like "Where is the Eiffel Tower?" or "Is the Eiffel Tower in Paris?" fall under this sub-task. Other factual knowledge, such as "The capital of the UK is London," can also be defined as sub-tasks. Hence, the ideas of model disassembly and reassembly have potential applications in NLP models, including large language models, and can be used for learning or knowledge editing.
>
> **On the Generalization of the Method**: The entire method is not overfitted to classification tasks and CNN architectures. Below are core analyses and explanations of our proposed MDA method:
>
> 1. Model Disassembly: Our model disassembly method is based on the proposed contribution aggregation and allocation. Contribution refers to the impact of features or parameters on the model's final prediction (lines 97-99 of the paper). Contribution aggregation refers to how contributions flow when multiple input neurons point to a single output neuron, and contribution allocation refers to how contributions flow when a single input neuron points to multiple output neurons (Figure 1 in Section 3.1 and Figure 5 in Appendix C). Based on contribution aggregation and allocation, we can continuously locate features most relevant to the prediction and thus identify the parameters most closely connected to these features (lines 157-162).
> 2. Model Reassembly: For disassembled parameters, through padding strategies to match parameter dimensions (lines 208-211) and parameter scaling strategies to align parameter magnitudes (lines 228-232), sub-models from different source models can be assembled and directly used for inference.
>
> As seen, our proposed model disassembly and assembly method is not strictly bound to CNNs. It can serve as a general paradigm for solving model disassembly and reassembly tasks, with the potential to generalize to other tasks and deep neural networks. Specifically, besides convolutional layers, we also provide methods for contribution aggregation and allocation in linear layers, which are essential for networks like graph neural networks (Appendix C). Furthermore, in Section 5.3, we demonstrate the effectiveness of our method on graph convolutional networks, indicating that the MDA method remains effective.  Additionally, in response to your third comment (Q3), we have included experimental results demonstrating the applicability of our proposed method to Transformer models.
>
> **On Applicability to Self-supervised Learning**: We acknowledge the importance of self-supervised learning but believe it does not limit the applicability of our model disassembly and reassembly method.
>
> 1. Self-supervised learning generally involves using pre-tasks like context prediction or contrastive learning to obtain pre-trained models, which are then applied to downstream tasks like classification, detection, segmentation, or question answering in NLP. Given our above discussion on sub-task definitions and method generalization, applying model disassembly and assembly to models based on self-supervised learning is feasible and practical.
> 2. Regarding pre-trained models obtained through self-supervised learning, we believe directly applying model disassembly and reassembly to pre-trained models might not be particularly practical since these models are typically used for broad feature learning and cannot be directly applied to specific downstream tasks. However, this does not preclude the applicability of model disassembly and reassembly. For instance, pre-trained models can be decomposed into parameters related to the pre-task and unrelated ones, facilitating model compression by removing parameters unrelated to the pre-task. As shown in Appendix J.2, this approach to model compression has potential benefits.
>
> Thank you again for your valuable feedback. Applying model disassembly and reassembly to broader areas, such as NLP, is a key focus of our ongoing and future research efforts.

---

> ### Author Response · Authors · 2024-08-06
> **Rebuttal by Authors [Q2]**
>
> > Q2: *There is a lack of systematic comparisons with other works, such as deep model reassembly[1], and the study only includes self-constructed specific tasks, making it hard to evaluate the technical soundness.*
> >
>
> Thank you for your valuable feedback. Here is our response:
>
> **Systematic Comparisons with Other Methods**: We apologize for any confusion regarding this matter. In lines 44 to 47 of the main text and lines 503 to 504 of Appendix B, we highlight the key differences between our work and other related studies. To provide a systematic comparison, we have specifically compared our method, MDA, with the Deep Model Reassembly (DeRy) [1], as you pointed out. This comparison covers aspects such as problem definition, methodology, and effectiveness. It is evident that MDA differs significantly from DeRy and offers clear advantages. For instance, MDA disassembles models into task-aware components that handle different subtasks vertically. These disassembled submodels can be directly used for inference, are interpretable, and can be assembled to perform various tasks.
>
> Moreover, compared to other related work, MDA has several distinctive features. For example, there is no need for predefined subcomponents during the training of the original model, and no additional learning modules are required during disassembly and reassembly. In summary, this work represents the first approach to model disassembly and reassembly related to subtasks, allowing for the flexible creation of new models in a manner akin to assembling with building blocks.
>
> |  | DeRy [1] | MDA [This Paper] |
> | --- | --- | --- |
> | Problem Definition | DeRy aims to partition different layers of models into equivalent sets and then reassemble layers from these sets. | MDA is the first to aim at disassembling a model into different task-aware components, with each component corresponding to one or more specific sub-tasks, which can be assembled to solve a larger task comprising these sub-tasks. |
> | Model Disassembly Method | DeRy uses covering set optimization to partition different layers of models into equivalent sets. To maintain these sets, each time a new model is partitioned, it requires calculating the "functional similarity" for relevant layers across all models, increasing unnecessary computational overhead. | MDA disassembles the model based on proposed contribution aggregation and allocation, independent of other models and solely related to the sub-tasks themselves. This direct approach significantly enhances disassembly efficiency and practicality. |
> | Effectiveness of Disassembled Models | The parameters generated by DeRy's method, which partitions models by layers, lack practical meaning, resembling a black box that cannot be understood or used directly for inference. | MDA disassembles the model according to the inference paths of different sub-tasks, making the model more transparent and interpretable. More importantly, the task-aware components disassembled can be used directly for inference without any training. |
> | Model Reassembly Method | DeRy reassembles models by solving integer programming to stack layers from different models horizontally. If the dimensions of two consecutive layers do not match, additional parameters for concatenation layers are introduced, necessitating retraining. | MDA reassembles models by vertically assembling different decision paths and uses a parameter scaling strategy that requires no training to mitigate sub-task interference. This approach usually yields excellent results and maintains interpretability during the assembly phase. |
> | Effectiveness of Reassembled Models | DeRy can only partition and reassemble models for the same task. For example, different models for ImageNet classification can be partitioned by layers and reassembled, but the new model can still only be used for ImageNet classification. | MDA can assemble models used for different sub-tasks. The reassembled model can solve new, previously unseen larger tasks comprising these sub-tasks. |
> |  |  |  |
>
> **Self-constructed Specific Tasks**: Here is our response to your concern:
>
> 1. This work is the first to propose sub-task-based deep model disassembly and assembly (Section 2). Therefore, there are currently no standard datasets or tasks, nor comparable settings from previous works.
> 2. The datasets used to evaluate the final model performance are publicly recognized, and the evaluation process is standard. In addition to the commonly used model accuracy, we provide more detailed experimental results, such as parameter counts and FLOPs of disassembled models (Appendix G) and more granular baseline effects before model assembly (Appendix H).
>
> We understand your concerns and hope that the above response addresses your questions. Designing a comprehensive benchmark for model disassembly and reassembly will be one of our priorities in future work.

---

> ### Author Response · Authors · 2024-08-06
> **Rebuttal by Authors [Q3&Q4]**
>
> > Q3: *Given the widespread adoption of transformers in the field, the absence of experiments involving transformers is unfortunate.*
> >
>
> We understand your concern. In fact, the proposed MDA, such as contribution aggregation and allocation, can be applied to transformers, as discussed in our response to your first comment (Q1). We conducted a preliminary model disassembly experiment using ViT-Tiny on ImageNet-10, and the results are shown in Table 1.
>
> Table 1: Disassembly Performance of ViT-Tiny on ImageNet-10
>
> | Disassembled Task | Base(%) | Disa(%) |
> | --- | --- | --- |
> | 0-1 | 87.70 | 89.25 (+1.55) |
> | 1-3 | 75.10 | 78.10 (+3.00) |
> | 3-9 | 77.51 | 79.29 (+1.78) |
>
> As shown in Table 1, the disassembled model can still be used for inference, and its accuracy is higher than that of the original model.
>
> Furthermore, we tested the model assembly performance of ViT-Tiny on CIFAR-10 and CIFAR-100. However, we must acknowledge that the results were not satisfactory. For example, when assembling the disassembled CIFAR-10 (0-3) and CIFAR-100 (11-17) models, the accuracy without fine-tuning was only 13.27%. As Reviewer gfoc mentioned in their final comment, task interference occurs when sub-models from different source models are assembled together, and this issue is particularly severe due to the self-attention mechanism unique to transformers. We believe that adjusting parameter distribution from different sub-model parameter spaces before assembly is a potential solution to this problem, which will be a focus of our future research.
>
> > Q4: *Could you provide an intuitive explanation for why your method sometimes outperforms the baseline, given that the baseline is end-to-end trained for the classification task?*
> >
>
> That's an excellent question. The superior performance of the proposed MDA method over the base model can be attributed to two main factors:
>
> 1. The source model, from which the disassembled components are derived, can be considered as trained on additional data relative to the target task. This means it has learned more features and possesses better feature extraction capabilities.
> 2. More critically, during model disassembly, our proposed method extracts only the parameters relevant to the subtasks, discarding those unrelated. This means that during inference, there is no interference from parameters associated with other subtasks, leading to higher inference accuracy.
>
> The performance improvement demonstrates that disassembling and identifying subtask-related model parameters is both feasible and effective. Besides the accuracy improvement, the experiments in our paper also show that our method accurately identifies subtask-related parameters and effectively removes irrelevant ones (as detailed in Table 5 of Appendix G).

---

> ### Author Response · Authors · 2024-08-11
> **Willing to Provide Further Clarifications**
>
> Dear Reviewer,
>
> Thank you for your valuable comments and feedback. We have thoroughly addressed the questions you raised in our detailed responses. If you have any further questions or concerns, we would be more than happy to provide additional clarifications. We believe this would be greatly beneficial in refining and improving our work.
>
> Sincerely,
> The Authors of Model Lego

---

> > ### Comment · Reviewer_UTXG · 2024-08-11
> >
> > Thank you for your detailed response.
> >
> > Although NeurIPS rules state that "Reviewers are not required to take comments into consideration," I reviewed all your comments. However, I found them not satisfactory.
> >
> > Specifically, the main issues with your response are as follows:
> >
> > The response was *overwhelmingly long*, exceeding the NeurIPS rebuttal limit of 6000 characters. This length made it difficult for me to grasp the key points and conclusions in your response.
> >
> > The overall content seemed somewhat vague, with extensive explanations of what your method *might* achieve. However, an experiment result is worth more than a thousand words. For instance, you spent considerable effort discussing whether your method could be applied to other tasks like generation or self-supervised learning, but the explanations remained highly theoretical and philosophical without supporting results. Moreover, I was hoping for more system level with other methods. I expected to see how your method outperforms others in a fair setting, demonstrating clear advantages or unique capabilities, but such results were absent.
> >
> > Additionally, your experiments applying the method to transformers further demonstrated that the generalizability of the approach is quite limited (even in a very toy setting, from my perspective). This reinforces the idea that philosophical explanations is far from insufficient in validating your claims. Practical experiments reveal that applying this method to other tasks or models is not as straightforward as suggested. So I would also expect similar difficulties in applying your method to other tasks (for example, masked image modeling or next-token-prediction in self-supervised learning).
> >
> > Given the above points, my concerns are not addressed with the author's response, and I will be maintaining my original rating.

---

> ### Author Response · Authors · 2024-08-13
>
> Dear Reviewer,
>
> Thank you for your constructive suggestions.
>
> We apologize for exceeding the word limit in our previous response. Here, we summarize our replies to your four comments on weaknesses and questions (denoted as Q1 to Q4) in the most concise manner:
>
> - **Limitation of Subtask Definitions (Q1)**: This work introduces model disassembly and reassembly related to subtasks (Section 2). Besides classification, it is evident that subtasks can be defined in image detection, segmentation, generation, and subtasks related to factual knowledge in natural language.
>
> - **Complexity and Generalization of the Method (Q1)**: The proposed approach is not coupled with the CNN architecture. This paradigm can be applied to a broader range of deep neural networks, such as graph neural networks (Section 5.3 in the paper).
>
> - **Applicability of Self-Supervised Learning (Q1)**: Regarding self-supervised learning, such as masked image modeling, the pre-trained model obtained is used for feature learning and is not tied to a specific task. Thus, only subtasks related to feature learning can be defined and utilized for model compression (as discussed in Appendix J.2). However, for downstream task models based on self-supervised learning, our method is equally applicable as it is to standard models.
>
> - **Systematic Comparison with Other Methods (Q2)**: This work is the first to propose model disassembly and assembly related to subtasks, significantly differing from existing work. For example, previous work does not even decompose a sub-model that can be directly inferred from. Therefore, fair quantitative comparisons are not feasible. Nevertheless, we provide detailed comparisons across problem definition, methods, and effects, highlighting the advantages of our approach.
>
> - **Specific Tasks of Self-Constructed Models (Q2)**: As the first work on model disassembly and assembly, there are no established experimental settings to follow. However, our experimental environment is fair, using publicly available datasets, with detailed experimental setups provided. In addition to model accuracy, we have also provided experimental data on model parameters and FLOPs (Appendix G), as well as more fine-grained baselines (Appendix H).
>
> - **Experimental Results on Transformers (Q3)**: We have supplemented experimental results on Transformers, demonstrating that disassembled sub-models perform well on different subtasks. However, due to the unique nature of self-attention mechanisms, there can be interference between subtasks when assembling these sub-models, leading to unsatisfactory performance without further fine-tuning. We have provided explanations and potential solutions for this issue. Along with model disassembly and reassembly on large language models, this will be a direction for our future work.
>
> - **Reasons for Performance Exceeding Baselines (Q4)**: The primary reason is that the original model, from which the components are disassembled, has been trained on more extensive data compared to the baseline models for the target tasks. Furthermore, only parameters relevant to subtasks are included in the disassembled model, reducing interference from other task parameters and thus leading to more accurate predictions.
>
> The detailed responses to these points can be found in the rebuttal above. Once again, we appreciate your thorough review and valuable suggestions.
>
> Best regards,
> The Model LEGO Authors

---

### Official Review · Reviewer_FhUd · 2024-07-14

**Soundness:** 4
**Presentation:** 3
**Contribution:** 4
**Rating:** 7
**Confidence:** 4

**Summary:**

The paper introduces Model Disassembling and Assembling (MDA), a novel method inspired by the biological visual system to create new deep learning models without retraining. By disassembling pretrained CNNs into task-aware components and reassembling them, the approach maintains performance while enabling efficient model reuse. Experiments show that the reassembled models match or exceed the original models' performance, highlighting MDA's potential for applications like model compression and knowledge distillation.

**Strengths:**

1. The proposed method allows for arbitrary assembly of new models from task-aware components, similar to building with LEGO blocks, which is novel and interesting.

2. MDA creates new models without requiring extensive retraining, saving significant computational resources.

3. The experimental results are impressive. The reassembled models can match or even surpass the performance of the original models.

**Weaknesses:**

1. The computation of contributions and component locating may introduce additional time and computational cost, especially when the number of classes or tasks are large. It is recommended to add a analysis of the complexity of this method.

2. Is this method applicable to transformer models and large models?

3. In some cases, the performance of MDA can even outperforms the base model, why?

4. Some typos in writing should be corrected. For example, in Section 5.2.1, the quotes `’sth’` are all right quotes.

**Questions:**

See Weaknesses.

**Limitations:**

The authors have adequately discussed the limitations.

---

> ### Author Rebuttal · Authors · 2024-08-06
>
> Thank you for your detailed review and comments on our work. We are pleased that you found our work to be novel and interesting, and that you found the experimental results impressive. We are also delighted that you pointed out how our method allows for the arbitrary creation of new models, similar to playing with Lego bricks, significantly saving computational resources. Below are our responses to each of your comments (your comments are highlighted in italics).
>
> > Q1: *The computation of contributions and component locating may introduce additional time and computational cost, especially when the number of classes or tasks are large. It is recommended to add a analysis of the complexity of this method.*
> >
>
> We greatly appreciate your concern regarding this issue. The proposed method indeed introduces additional computational costs. Specifically, let $K$ represent the number of classes or tasks, and $d$ represent the total dimension of features in the network. The complexity of our method consists of the following components:
>
> - Contribution Aggregation and Allocation: The computational complexity for this part is $O(K \cdot d)$.
> - Component Locating: The computational complexity for this part is $O(K \cdot d)$.
>
> Although the proposed method introduces the aforementioned computational costs, these steps are essential for accurately identifying the parameters relevant to the subtasks. Moreover, compared to training a model, the time overhead required for these steps is almost negligible.
>
> > Q2: *Is this method applicable to transformer models and large models?*
> >
>
> Thank you for your question. Here is our response:
>
> **Applicability to Transformer Models**: The proposed MDA method, including the underlying concepts of contribution aggregation and allocation, is indeed applicable to Transformer models. The method is not tightly coupled with CNNs and can be applied to linear layers and other structures beyond convolutional layers (Appendix C), making it a general paradigm for model disassembly and reassembly tasks. This is demonstrated in the experiments presented in Section 5.3 of the paper. Additionally, we conducted a preliminary model disassembly experiment using ViT-Tiny on ImageNet-10, and the results are shown in Table 1.
>
> Table 1: Disassembly performance of ViT-Tiny on ImageNet-10.
>
> | Disassembled Task | Base(%) | Disa(%) |
> | --- | --- | --- |
> | 0-1 | 87.70 | 89.25 (+1.55) |
> | 1-3 | 75.10 | 78.10 (+3.00) |
> | 3-9 | 77.51 | 79.29 (+1.78) |
>
> As shown in Table 1, the disassembled model remains functional for inference, with accuracy surpassing that of the original model.
>
> Additionally, we tested the assembly performance of ViT-Tiny on CIFAR-10 and CIFAR-100. However, the results were not satisfactory. For example, when assembling disassembled CIFAR-10(0-3) and CIFAR-100(11-17) models, the accuracy without fine-tuning was only 13.27%. The primary reason for this is the task interference that occurs when submodels from different source models are assembled, a phenomenon that is particularly severe due to the self-attention mechanism inherent in Transformers. We believe that adjusting parameter distribution from different submodel parameter spaces before assembly could be a potential solution to this issue, which will be a focus of our future work.
>
> **Applicability to Large Models**: Applying MDA to large models, such as GPT models based on decoder-only Transformers, is feasible and is a current and future focus of our research. For large models in tasks like natural language generation or question answering, it is crucial to clearly define subtasks. Our current approach involves defining different factual knowledge (e.g., "The Eiffel Tower is located in Paris" and "The first Olympic Games were held in Athens") as subtasks. Architecturally, the primary parameter layers in Transformers are linear layers, and our MDA method's contribution aggregation and allocation are applicable to these layers. The main challenge lies in addressing the task interference issue mentioned earlier.
>
> In conclusion, we fully agree on the importance of the question you raised. We believe that the proposed MDA method can serve as a general paradigm applicable to various neural networks, and advancing the disassembly and reassembly of models on Transformers and large Transformer-based models is a key focus of our ongoing and future research efforts.
>
> > Q3: *In some cases, the performance of MDA can even outperforms the base model, why?*
> >
>
> This is an excellent question. The superior performance of the proposed MDA method over the base model can be attributed to two main factors:
>
> 1. The source model, from which the disassembled components are derived, can be considered as trained on additional data relative to the target task. This means it has learned more features and possesses better feature extraction capabilities.
> 2. More critically, during model disassembly, our proposed method extracts only the parameters relevant to the subtasks, discarding those unrelated. This means that during inference, there is no interference from parameters associated with other subtasks, leading to higher inference accuracy.
>
> The performance improvement demonstrates that disassembling and identifying subtask-related model parameters is both feasible and effective. Besides the accuracy improvement, the experiments in our paper also show that our method accurately identifies subtask-related parameters and effectively removes irrelevant ones (as detailed in Table 5 of Appendix G).
>
> > Q4: *Some typos in writing should be corrected. For example, in Section 5.2.1, the quotes `’sth’` are all right quotes.*
> >
>
> Thanks for pointing it out. We have corrected the typos in the paper and thoroughly reviewed the entire document.

---

### Author Rebuttal · Authors · 2024-08-06

Dear Reviewers FhUd, UTXG, gfoc, and 17d9,

Thank you for the time and effort you have invested in reviewing our paper. We are particularly grateful for your recognition of the novelty and insight of our work and are pleased that our proposed model disassembly and reassembly approach has been deemed inspiring for model interpretability and model design.

Through our detailed responses to each of your questions and comments, we believe that the paper has been significantly improved. We have compiled the common concerns raised by the reviewers and have addressed each comment individually. If you find that our responses satisfactorily address your concerns, we would be grateful if you could consider increasing the score. If you have any further questions, we would be more than happy to engage in additional discussions.

> Reviewers FhUd, UTXG, gfoc, and 17d9 all raised questions about why the performance of MDA can even exceed that of the base model.
>

We apologize for any confusion caused. We have addressed this question in detail in the responses to each reviewer’s comments. In summary, the performance improvement of MDA can be attributed to two main factors: 1.The original models from which the components are disassembled can be considered as having been trained on additional data compared to the baseline model for the target tasks. 2.The disassembled submodels contain only the parameters relevant to the corresponding subtasks, which helps avoid interference from parameters related to other subtasks.

> Reviewers FhUd, UTXG, and 17d9 all raised questions about the applicability of MDA beyond convolutional neural networks.
>

We have addressed this question in detail in our responses to each reviewer. In summary, we have explained the potential applications of MDA methods to other architectures and tasks, such as classification, segmentation, and large language models. Additionally, we have provided supplementary experiments in the paper and responses demonstrating the effectiveness of MDA in architectures like graph convolutional networks and Transformers.

While model disassembly is applicable to networks such as Transformers, we acknowledge that task interference can occur during model assembly in Transformer architectures. We have provided explanations for this phenomenon and potential solutions. We will continue to explore the applications of model disassembly and assembly in deep learning, with a focus on Transformer models and Transformer-based large language models as a key area of ongoing and future research.

> Reviewer UTXG has focused on issues related to the definition of subtasks in model disassembly and reassembly, the generalizability of the method, the applicability of self-supervised learning, and a systematic comparison with related methods.
>

Thank you for your comments. We have provided detailed responses to each of these issues under the respective comments. These include explanations on the definition of subtasks related to images and non-images, an overview of the proposed method and its generalizability, how model disassembly and reassembly can be applied to self-supervised learning, and a systematic comparison with other methods. We believe these detailed explanations will help reduce any misunderstandings about our paper.

> Reviewer gfoc emphasized the issue of category or subtask interference during model assembly in several comments.
>

Thank you for your suggestions. We have provided detailed explanations under the respective comments regarding the causes and consequences of interference between classes or subtasks, the specific measures we have taken to mitigate this interference, and potential future approaches to further address it. We believe these responses will effectively address your concerns.

Beyond the issues mentioned above, more detailed responses to each reviewer’s comments can be found below their respective sections. We sincerely thank the reviewers for their time and valuable feedback, which have greatly contributed to the improvement of the paper.

---

### Comment · Area_Chair_XtJa · 2024-08-11
**Discussion Period**

Dear Reviewers,

Please read the authors' rebuttal and post your response to the rebuttal. Thank you.

Best,
AC

---

### Decision · Program_Chairs · 2024-09-25

**Decision:**

Accept (poster)

**Comment:**

The submission received three Accept (7) and one Strong Reject (2). Reviewer UTXG raised concerns including complicated method, limiting to classification tasks, lack of systematic comparisons with "Deep model reassembly.", the absence of transformer-based experiments. The AC checked the authors' rebuttal to Reviewer UTXG and was convinced that these concerns are well addressed. All the reviewers agreed the **technical novelty of the paper, including Reviewer UTXG**. Given the strong recommendation from three reviewers and the technical novelty of the paper,  the AC recommends this paper for acceptance.